

**A review of experimental techniques for aerosol hygroscopicity studies**
Mingjin Tang,[1,*] Chak K Chan,[2,*] Yong Jie Li,[3] Hang Su,[4,5] Qingxin Ma,[6] Zhijun Wu,[7] Guohua
Zhang,[1] Zhe Wang,[8] Maofa Ge,[9] Min Hu,[7] Hong He,[6,10,11] Xinming Wang[1,10,11]
[1] State Key Laboratory of Organic Geochemistry and Guangdong Key Laboratory of
Environmental Protection and Resources Utilization, Guangzhou Institute of Geochemistry,
Chinese Academy of Sciences, Guangzhou 510640, China
[2] School of Energy and Environment, City University of Hong Kong, Kowloon, Hong Kong,
China
[3] Department of Civil and Environmental Engineering, Faculty of Science and Technology,
University of Macau, Avenida da Universidade, Taipa, Macau, China
[4] Center for Air Pollution and Climate Change Research (APCC), Institute for Environmental
and Climate Research (ECI), Jinan University, Guangzhou 511443, China
[5] Department of Multiphase Chemistry, Max Planck Institute for Chemistry, Mainz 55118,
Germany
[6] State Key Joint Laboratory of Environment Simulation and Pollution Control, Research
Center for Eco-Environmental Sciences, Chinese Academy of Sciences, Beijing 100085, China
[7] State Key Joint Laboratory of Environmental Simulation and Pollution Control, College of
Environmental Sciences and Engineering, Peking University, Beijing 100871, China
[8] Department of Civil and Environmental Engineering, The Hong Kong Polytechnic University,
Hong Kong, China
[9] State Key Laboratory for Structural Chemistry of Unstable and Stable Species, Institute of
Chemistry, Chinese Academy of Sciences, Beijing 100190, China
[10] University of Chinese Academy of Sciences, Beijing 100049, China



[11]  Center for Excellence in Regional Atmospheric Environment, Institute of Urban
Environment, Chinese Academy of Sciences, Xiamen 361021, China
Correspondence:    Mingjin    Tang    (mingjintang@gig.ac.cn),    Chak    K.    Chan
(Chak.K.Chan@cityu.edu.hk)
**Abstract**

Hygroscopicity is one of the most important physicochemical properties of aerosol

particles, and also plays indispensable roles in many other scientific and technical fields. A
myriad of experimental techniques, which differ in principles, configurations and cost, are
available for investigating aerosol hygroscopicity under subsaturated conditions (i.e., relative
humidity below 100%). A comprehensive review of these techniques is provided in this paper,
in which experimental techniques are broadly classified into four categories, according to the
way samples under investigation are prepared. For each technique, we describe its operation
principle and typical configuration, use representative examples reported in previous work to
illustrate how this technique can help better understand aerosol hygroscopicity, and discuss its
advantages and disadvantages. In addition, future directions are outlined and discussed for
further technical improvement and instrumental development.



## 1 Introduction

Aerosol particles are airborne solid or liquid particles in the size range of a few nanometers to tens of micrometers. They can be emitted directly into the atmosphere (primary particles), and can also be formed in the atmosphere (secondary particles) by chemical transformation of gaseous precursors such as $SO_2$, NOx, and volatile organic compounds (Pöschl, 2005; Seinfeld and Pandis, 2016). Aerosol particles are of great concerns due to their environmental, health, climatic and biogeochemical impacts (Finlayson-Pitts and Pitts, 2000; Jickells et al., 2005; Mahowald, 2011; Mahowald et al., 2011; IPCC, 2013; Pöschl and Shiraiwa, 2015; Seinfeld and Pandis, 2016; Shiraiwa et al., 2017b).

Water, which can exist in gas, liquid and solid states, is ubiquitous in the troposphere. Interactions of water vapor with aerosol particles largely affect the roles that aerosol particles play in the Earth system. When water vapor is supersaturated (i.e. when relative humidity, RH, is >100%), aerosol particles can act as cloud condensation nuclei (CCN) to form cloud droplets and as ice-nucleating particles (INPs) to form ice crystals (Pruppacher and Klett, 1997; Lohmann and Feichter, 2005; Vali et al., 2015; Lohmann et al., 2016; Tang et al., 2016a; Knopf et al., 2018; Tang et al., 2018). Cloud condensation nucleation and ice nucleation activities of aerosol particles, as well as relevant experimental techniques, have been recently reviewed in several books and review papers (Pruppacher and Klett, 1997; Hoose and Moehler, 2012; Murray et al., 2012; Kreidenweis and Asa-Awuku, 2014; Farmer et al., 2015; Lohmann et al., 2016; Tang et al., 2016a; Kanji et al., 2017), and are thus not further discussed in this paper.

When water vapor is unsaturated (i.e. RH <100%), an aerosol particle in equilibrium with the surrounding environment would contain some amount of absorbed or adsorbed water (Martin, 2000; Kreidenweis and Asa-Awuku, 2014; Cheng et al., 2015; Farmer et al., 2015; Seinfeld and Pandis, 2016; Tang et al., 2016a; Freedman, 2017). The amount of water that a particle contains depends on RH, temperature, its chemical composition, size, and etc. The



ability of a substance to absorb/adsorb water as a function of RH is typically termed as
hygroscopicity (Adams and Merz, 1929; Su et al., 2010; Kreidenweis and Asa-Awuku, 2014;
Tang et al., 2016a), and the underlying thermodynamic principles can be found elsewhere
(Martin, 2000; Seinfeld and Pandis, 2016). A single-component particle which contains one of
water soluble inorganic salts, such as $(NH_4)_2SO_4$ and $NaCl$, is solid at low RH. When RH is
increased to the deliquescence relative humidity (DRH), the solid particle will undergo
deliquescence to form an aqueous particle, and the aqueous particle at DRH is composed of a
saturated solution (Cheng et al., 2015). Further increase in RH would increase the water content
of the aqueous droplet, i.e. the aqueous particle would become more diluted as RH increases.
During humidification thermodynamics determines phase transition and hygroscopic growth
of the particle. During dehumification, an aqueous particle would not undergo efflorescence to
form a solid particle when RH is decreased to below DRH; instead, the aqueous particle would
become supersaturated. Only when RH is further decreased to efflorescence relative humidity
(ERH), the aqueous particle would undergo crystallization, leading to the formation of a solid
particle. Therefore, efflorescence is also kinetically controlled and there is a hysteresis between
deliquescence and efflorescence. Deliquescence and efflorescence of multicomponent particles
can be more complicated (Seinfeld and Pandis, 2016).

It should be pointed out that not all the single-component particles exhibit distinctive

deliquescence and efflorescence. Instead, continuous uptake or loss of liquid water is observed
during humidification and dehumidification processes for many inorganic and organic particles
(Mikhailov et al., 2009; Koop et al., 2011; Shiraiwa et al., 2011). Particles with extremely low
hygroscopicity (e.g., mineral dust) will not be deliquesced even at very high RH; instead,
adsorbed water will be formed on the particle surface (Tang et al., 2016a). Furthermore, a
multicomponent particle which contains some types of organic materials may undergo liquid-
liquid phase separation, leading to the formation two coexisting liquid phases in one particles



(Mikhailov et al., 2009; You et al., 2012; You et al., 2014; Freedman, 2017; Song et al., 2017;
Song et al., 2018). It is conventionally assumed that hygroscopic equilibrium of aerosol
particles can be quickly reached. Nevertheless, recent laboratory, field and modeling studies
suggested that atmospherically relevant particles can be semi-solid or amorphous solid
(Virtanen et al., 2010; Zobrist et al., 2011; Renbaum-Wolff et al., 2013; Shiraiwa et al., 2017a;
Reid et al., 2018). The viscosity of these particles can be high enough such that uptake or
release of water is largely limited by diffusion of water molecules in the bulk phase of these
particles.
Hygroscopicity determines the amount of water that a particle contains under a given
condition and thereby has several important implications. It affects the size and refractive
indices of aerosol particles, affecting their optical properties and consequently their impacts on
visibility and direct radiative forcing (Malm and Day, 2001; Chin et al., 2002; Quinn et al.,
2005; Hand and Malm, 2007; Cheng et al., 2008; Eichler et al., 2008; Liu et al., 2012; Liu et
al., 2013b; Brock et al., 2016b; Titos et al., 2016; Haarig et al., 2017). Hygroscopicity is also
closely related to the CCN activity of aerosol particles, affecting their impacts on formation
and properties of clouds and thus their indirect radiative forcing (McFiggans et al., 2006;
Petters and Kreidenweis, 2007; Reutter et al., 2009; Kreidenweis and Asa-Awuku, 2014;
Farmer et al., 2015). Aerosol liquid water and/or surface-adsorbed water, largely controlled by
hygroscopicity, determines heterogeneous and multiphase reactions of aerosol particles via
several mechanisms, as revealed by recent studies (Bertram and Thornton, 2009; Shiraiwa et
al., 2011; Rubasinghege and Grassian, 2013; Cheng et al., 2016; Wang et al., 2016; Tang et al.,
2017; Mu et al., 2018; Wu et al., 2018). In addition, hygroscopicity significantly impacts dry
and wet deposition rates of aerosol particles and thus their lifetimes, spatiotemporal distribution
and environmental and health effects (Fan et al., 2004; Wang et al., 2014a). For primary
biological aerosols in specific, changes in their atmospheric transport behavior have important



implications for the spread of plants and microbes and therefore the evolution of ecosystems
(Brown and Hovmoller, 2002; Després et al., 2012; Fisher et al., 2012; Fröhlich-Nowoisky et
al., 2016).
Atmospheric aerosol is only one of many fields in which hygroscopicity is of great interest.
Hygroscopicity is closely linked to water activities and thermodynamics of solutions (Atkins,
1998). It also determines the amount of surface-adsorbed water and surface reactivity of
various solid materials, and has been widely investigated in surface science and heterogeneous
catalysis (Miranda et al., 1998; Ewing, 2006; Yamamoto et al., 2010b; Chen et al., 2012;
Rubasinghege and Grassian, 2013; Liu et al., 2017). Hygroscopicity is related to the possible
existence of liquid water in some hyperarid environments (such as Mars and the Atacama
Desert on earth) (Martin-Torres et al., 2015): while pure liquid water is not stable in these
environments, the deliquescence of some salts, such as chlorides and perchlorates, can occur
at RH significantly below 100% and lead to the formation of aqueous solutions (Gough et al.,
2011; Gough et al., 2016; Gu et al., 2017a; Jia et al., 2018). Hygroscopic properties
significantly affect transport and deposition of inhaled aerosol particles in the respiratory tract,
therefore playing an important role in the health impact of ambient aerosols as well as efficacy
and side effects of aerosolized pharmaceuticals (Hickey and Martonen, 1993; Robinson and
Yu, 1998; Carvalho et al., 2011; Hofmann, 2011; Haddrell et al., 2014; Winkler-Heil et al.,
2014; Darquenne et al., 2016; Davidson et al., 2017; Winkler-Heil et al., 2017). Impacts of
moisture and implications of hygroscopicity have been well documented for physical and
chemical stability of pharmaceuticals (Ahlneck and Zografi, 1990; Chan et al., 1997; Peng et
al., 2000; Newman et al., 2008; Mauer and Taylor, 2010b; Tong et al., 2010a; Feth et al., 2011)
as well as food ingredients and blends (Mauer and Taylor, 2010a; Allan and Mauer, 2016), and
large efforts have been made in pharmaceutical and food industry to prevent relevant products
from deliquescence. Corrosion and degradation of various constructions and buildings depend



largely on RH, and as a result both chemical inertness and hygroscopicity of materials used
should be taken into account (Schindelholz et al., 2014a; Schindelholz et al., 2014b; Vainio et
al., 2016); in addition, deposition of particles of different compositions has also been shown to
affect the extent of corrosion of mild steel (Lau et al., 2008).

As summarized in this paper, a number of experimental techniques, which differ largely

in principles, configurations and cost, have been developed to investigate hygroscopic
properties of atmospherically relevant particles. Hygroscopic properties investigated at <100%
RH typically include the amount of water absorbed/adsorbed by particles as a function of RH,
as well as DRH and ERH if they exist. Techniques employed to investigate aerosol
hygroscopicity under supersaturation, commonly termed as CCN activity, are relatively less
diverse, and interested readers are referred to relevant literature (Nenes et al., 2001; Roberts
and Nenes, 2005; Kreidenweis and Asa-Awuku, 2014) for further information. In addition,
technique used to study ice nucleation have been discussed in a number of recent papers
(DeMott et al., 2011; Murray et al., 2012; Ladino et al., 2013; DeMott et al., 2018) and as a
result are not further discussed here.

Several review papers and book chapters have discussed some of these techniques used to

investigate aerosol hygroscopicity. For example, Kreidenweis and Asa-Awuku (2014)
discussed a few widely used techniques for aerosol hygroscopicity measurements, and Tang et
al. (2016) summarized in brief experimental techniques used to investigate water adsorption
and hygroscopicity of mineral dust particles. There are also a few review papers focused on a
specific technique or a specific category of techniques. For example, Swietlicki et al. (2008)
reviewed aerosol hygroscopicity measured in various environments using humidity-tandem
differential mobility analysers, and provided a nice overview of this technique; application of
single particle levitation techniques to investigate properties and processes of aerosol particles,
including aerosol hygroscopicity, was reviewed by Krieger et al. (2012); Titos et al. (2016)



reviewed techniques used to investigate the effect of hygroscopic growth on aerosol light
scattering, and Ault and Axson (2017) summarized and discussed recent advancements in
spectroscopic and microscopic methods for characterization of aerosol composition and
physicochemical properties.
Nevertheless, to our knowledge there is hitherto no paper or book which covers most of
(if not all) experimental techniques used for hygroscopicity measurements. This paper aims at
providing the first comprehensive review in this field. For each technique, we first introduce
its operation principle and typical configurations, and then use exemplary results to illustrate
how this technique can help better understand hygroscopic properties. According to the way
samples under investigation are prepared, experimental techniques covered in this paper are
classified into four categories, which are discussed in Sections 2-5. In Section 2, we discuss
experimental techniques applied to bulk solutions. Experimental techniques for particles
deposited on substrates, levitated single particles and aerosol particles are reviewed in Sections
3-5, respectively. Remote sensing techniques can also be employed to retrieve aerosol
hygroscopicity (Ferrare et al., 1998; Feingold and Morley, 2003; Pahlow et al., 2006; Schuster
et al., 2009; Li et al., 2013; Lv et al., 2017; Bedoya-Velasquez et al., 2018; Fernandez et al.,
2018); however, they are not covered in this paper because we intend to focus on in-situ
techniques and application of remote sensing to investigate aerosol hygroscopicity has been
discussed very recently in a book chapter (Kreidenweis and Asa-Awuku, 2014). In addition,
techniques for measuring CCN and IN activities of aerosol particles are not covered in the
present paper, and interested readers are referred to relevant literature (Roberts and Nenes,
2005; Lance et al., 2006; Petters et al., 2007; Good et al., 2010a; DeMott et al., 2011; Lathem
and Nenes, 2011; Hiranuma et al., 2015; Wex et al., 2015).



## 2 Bulk solution-based techniques

In principle, the hygroscopicity of a compound can be determined by measuring the water vapor pressure of air over (i.e. in equilibrium with) the aqueous solution at a given concentration (Pitzer, 1991; Rard and Clegg, 1997). Experimental data can then be used to derive water-to-solute ratios as a function of RH for aqueous solutions, and the RH over the saturated solution can generally be regarded as the DRH. Experimental methods based on this principle have been widely used since the early 20[th] century (or probably even earlier) (Adams and Merz, 1929; Hepburn, 1932) and are still being used (Königsberger et al., 2007; Sadeghi and Shahebrahimi, 2011; Golabiazar and Sadeghi, 2014) to investigate thermodynamic properties of aqueous solutions. In general, these methods can be further classified to two categories, i.e. isopiestic and nonisopiestic methods (Rard and Clegg, 1997).

### 2.1 The isopiestic method

The isopiestic method was described in a number of previous studies (Spedding et al., 1976; Rard and Miller, 1981; Pitzer, 1991; Hefter et al., 1997; Rard and Clegg, 1997; Königsberger et al., 2007), and a brief introduction is provided herein. For a typical experiment, two open vessels which contain a reference solution and a sample solution are housed in a sealed chamber with temperature being well controlled, and water vapor will be transferred between the two solutions until an equilibrium is reached. For the reference solution, its water activity should be well documented as a function of concentration. When the equilibrium is reached, the water activity of the sample solution is equal to that of the reference solution. If we measure the concentrations of the two solutions in equilibrium, the water activity of the sample solution at a given concentration can then be determined.

### 2.2 Nonisopiestic techniques

The water vapor pressure over (or the water activity of) an aqueous solution can be determined using a number of methods (Rard and Clegg, 1997), including but not limited to (i)




the static vapor pressure method, i.e. direct measurement of the vapor pressure over a solution
after being degassed (Adams and Merz, 1929; Jakli and Vanhook, 1972; Apelblat, 1992); (ii)
the dynamic vapor pressure method, i.e. measurements of the amount of water vapor from an
aqueous solution required to saturate a given volume of air (Bechtold and Newton, 1940); (iii)
measurements of the boiling temperature of an aqueous solution; (iv) measurements of the dew
point or RH of the air over an aqueous solution (Hepburn, 1932); and (v) the vapor pressure
osmometry (Amdur, 1974; Sadeghi and Shahebrahimi, 2011). These techniques are described
elsewhere (Pitzer, 1991; Rard and Clegg, 1997), and interested readers are referred to the two
papers (and references therein) for more information. A few recent studies are discussed below
to illustrate how nonisopiestic techniques could be used to investigate hygroscopic properties
of compounds relevant for atmospheric aerosols.

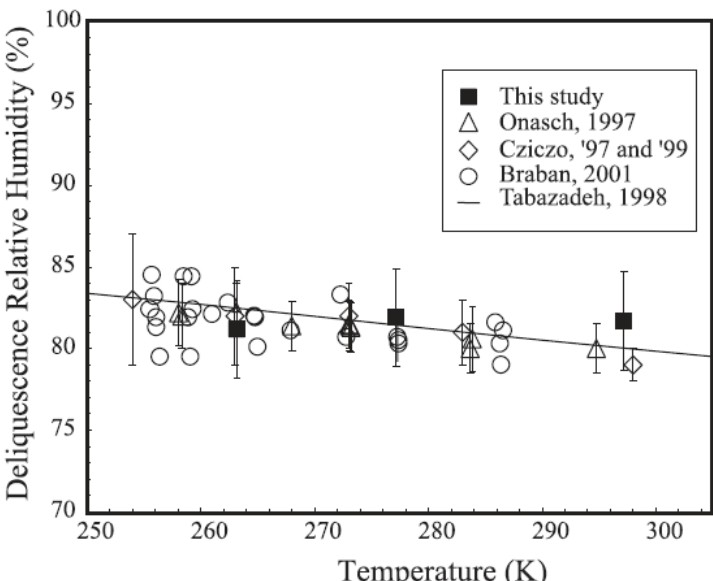


**Figure 1.** Comparison of DRH values as a function of temperature (250-300 K) measured by
different studies. Reprint with permission by Brook et al. (2002). Copyright 2002 John Wiley
& Sons, Inc.



235 The RH of air over 10 mL aqueous solutions which were contained in sealed test tubes

236 kept at constant temperatures were measured by Tolbert and coworkers (Brooks et al., 2002;

237 Wise et al., 2003) to investigate water activities as a function of solution concentration. In the

238 first study (Brooks et al., 2002), RH over saturated solutions were measured for $(NH_4)_2SO_4$,

239 several dicarboxylic acids, as well as mixtures of $(NH_4)_2SO_4$ with individual dicarboxylic acids

240 to determine their DRH. As shown in Fig. 1, the DRH values of $(NH_4)_2SO_4$ measured by

241 Brooks et al. (2002) agreed well with those reported in previous studies (Cziczo et al., 1997;

242 Tabazadeh and Toon, 1998; Cziczo and Abbatt, 1999; Onasch et al., 1999; Braban et al., 2001)

243 for temperature ranging from ~250 to ~300 K, confirming that the simple technique could

244 determine DRH in a reliable manner. It was further found that the presence of water soluble

245 dicarboxylic acids would reduce the DRH of $(NH_4)_2SO_4$, whereas the presence of less soluble

246 dicarboxylic acids had no measurable effects (Brooks et al., 2002). In a following study (Wise

247 et al., 2003), RH of air over eutonic mixtures of $(NH_4)_2SO_4$/dicarboxylic acids were measured

248 at 25 ºC to investigate the effect of organic acids on hygroscopic growth of $(NH_4)_2SO_4$. The

249 presence of water soluble dicarboxylic acids reduced hygroscopic growth of $(NH_4)_2SO_4$, while

250 the effect of less soluble dicarboxylic acids were found to be negligible (Wise et al., 2003).

251 Water activity meters, which measure the dew point temperature of the air in equilibrium

252 with an aqueous sample, are commercially available (Maffia and Meirelles, 2001; Marcolli et

253 al., 2004; Salcedo, 2006). For example, water activities meters were employed by Salcedo

254 (2006) and Maffia and Meirelles (2001) to study hygroscopic properties of organic acids and

255 their mixtures with $(NH_4)_2SO_4$ and $NH_4HSO_4$ at 25 ºC.

256 **2.3 Discussion**

257 Bulk solution-based techniques have the advantage of being inherently accurate and very

258 simple, while one major drawback is that these measurement cycles can be very time-

259 consuming, typically taking days up to months to reach the equilibrium (Königsberger et al.,



2007). Particle water content can be quantitatively determined for unsaturated solutions,
whereas no information can be provided for supersaturated solutions. Bulk solution-based
methods do not require particle sphericity assumption to derive particle water content, but
cannot be used to study water adsorption. Generally speaking, while these techniques are useful
for understanding properties of deliquesced particles, they are not applicable for direct
measurements of ambient aerosol particles.
**3 Particles deposited on substrates**

In this section we review and discuss techniques which can be used to investigate

hygroscopic properties of particles (either particle ensembles or individual particles) deposited
on substrates. This section is further divided to five parts: techniques for which changes in
water vapor and particle mass are measured to investigate particle hygroscopicity are reviewed
in Sections 3.1 and 3.2, and microscopic and spectroscopic tools employed to investigate
particle hygroscopicity are reviewed in Sections 3.3 and 3.4. Measurements of change in
electrical conductivity for understanding hygroscopic properties of particles are briefly
discussed in Section 3.5.
**3.1 Measurement of water vapor**

Particles would absorb/adsorb water vapor from the gas phase to reach a new equilibrium

as RH increases, while water vapor will be released if RH decreases. Measurement of change
in water vapor can be used to investigate hygroscopic properties. Exposure of water vapor to
particles can be achieved either in a static cell or in a flow cell.
**3.1.1 Physisorption analyser**

When exposed to water vapor, particles will absorb/absorb water vapor, leading to

depletion of water vapor in the system. The amount of water absorbed/adsorbed by particles
can be determined from the measured change in water vapor pressure (if the volume remains
constant), and the RH can be calculated from the final water vapor pressure when the



equilibrium is reached. The amount of water associated with particles can be determined as a
function of RH by varying the initial water vapor pressure.

Commercial instruments, usually designed to measure the Brunauer-Emmett-Teller (BET)

surface areas using nitrogen or helium (Torrent et al., 1990), have been utilized to investigate
hygroscopic properties of atmospherically relevant particles (Ma et al., 2010b; Ma et al., 2012b;
Hung et al., 2015). For example, Ma et al. (2010b) integrated an AUTOSORB-1-C instrument
(Quantachrome, US) with a water vapor generator, and employed this apparatus to investigate
hygroscopic properties of NaCl, $NH_4NO_3$ and $(NH_4)_2SO_4$. The measured DRH values and mass
hygroscopic factors were found to agree very well with those reported in literature (Ma et al.,
2010b). This method has proved to be very sensitive; as shown in Fig. 2, change in adsorbed
water as small as <0.5 monolayer can be reliably quantified (Ma et al., 2013a). In addition to
$CaSO_4$ and gypsum, this instrument was also employed to investigate hygroscopic properties
of fresh and aged $Al_2O_3$, MgO and $CaCO_3$ particles (Ma et al., 2012a).

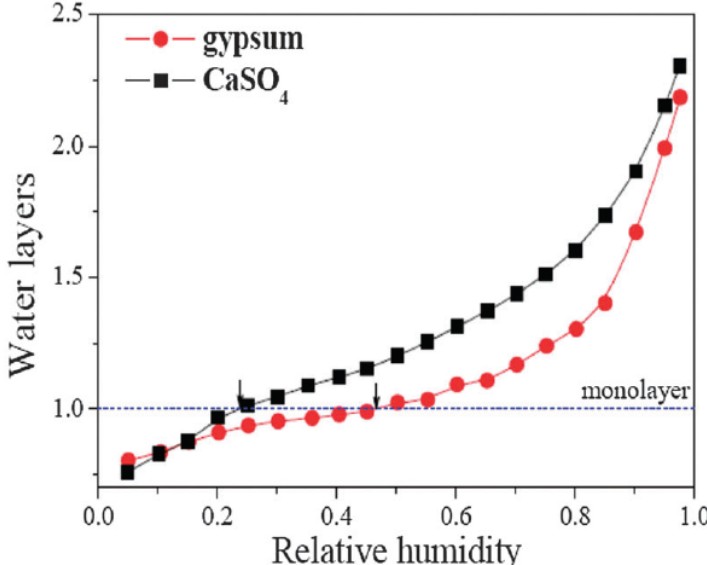




**Figure 2.** Water adsorption isotherms of $CaSO_4$ (black square) and gypsum ($CaSO_4 \cdot 2H_2O$, red
circle) at 278 K. Reprinted with permission by Ma et al. (2013a). Copyright 2013 the PCCP
Owner Societies.

A similar instrument (Micromeritics ASAP 2020) was employed by Hung et al. (2015) to

examine the hygroscopicity of black carbon, kaolinite and montmorillonite particles at 301 K,
and a sensitivity of sub-monolayers of adsorbed water could be achieved. Assuming a dry
particle diameter of 200 nm, the single hygroscopicity parameters, $\kappa$, were determined to be
~0.002 for montmorillonite and <0.001 for both black carbon and kaolinite (Hung et al., 2015).

This technique is able to quantify particle water content for unsaturated samples, and is

sensitive enough to measure adsorbed water; however, it cannot be (at least has not been) used
to examine supersaturated samples. This technique, which is independent on particle size and
morphology, can also been used to investigate hygroscopic properties of ambient aerosol
particles in an offline manner. For example, a physisorption analyser was used to study
hygroscopic properties of ambient aerosol particles collected in Beijing during an Asian dust
storm, and one monolayer of adsorbed water was formed on these particles at 46% RH (Ma et
al., 2012b).
**3.1.2 Katharometer**

The katharometer, also known as the thermal conductivity detector, can be used to measure

water vapor concentration. Lee and co-workers employed a katharometer to investigate liquid
water content of aerosol particles collected on filters (Lee and Hsu, 1998; Lee and Hsu, 2000;
Lee and Chang, 2002). In this setup (Lee and Chang, 2002), aerosol particles were collected
on a Teflon filter and then equilibrated with a helium flow at a given RH; after the equilibrium
was reached, the particle-loaded filter was purged with a dry helium flow, which was
subsequently directed to a katharometer to measure the water vapor concentration. As a result,



the liquid water content associated with particles at a given RH could be quantified. The
performance of this new method was systematically examined (Lee and Hsu, 1998; Lee and
Hsu, 2000; Lee and Chang, 2002), and the measured water-to-solute ratios at different RH
during both humidification and dehumidification processes were found to agree well with those
reported in literature for several compounds, including NaCl, NH$_4$Cl, Na$_2$SO$_4$, (NH$_4$)$_2$SO$_4$ and
NH$_4$NO$_3$.
Mikhailov et al. (2011, 2013) also developed a katharometer-based method to investigate
aerosol hygroscopicity. The instrument, called filter-based differential hygroscopicity analyser
(FDHA), are described elsewhere (Mikhailov et al., 2011), and a brief introduction is provided
here. In this apparatus, a humidified helium flow was split to two identical flows which were
then passed through a pair of differential measurement cells: the reference cell contained a
blank filter, and the sample cell contained a filter laden with particles (typically less than 0.1
mg). The difference in water vapor concentrations in these two cells, caused by
absorption/adsorption of water by particles loaded on the filter, was measured using a
differential katharometer, and the amount of water taken up by particles could be quantified by
integration of the katharometer signals over time. This instrument could measure hygroscopic
growth at very high RH (up to 99%).
Hygroscopic properties of (NH$_4$)$_2$SO$_4$, NaCl, levoglucosan, malonic acid, and mixed
(NH$_4$)$_2$SO$_4$/malonic acid particles were examined using FDHA at different RH during
humidification and dehumidification (Mikhailov et al., 2013), and the measured mass growth
factors agreed well with those reported in literature. This instrument was further employed to
investigate hygroscopic properties of particles collected from a pristine tropical rainforest (near
Manaus, Brazil) (Mikhailov et al., 2013), a suburban boreal forest site (near the city of St.
Petersburg, Russia) (Mikhailov et al., 2013) and a remote boreal site (the Zotino Tall Tower
Observatory, ZOTTO) in Siberia (Mikhailov et al., 2015). Fig. 3 displays the measured





hygroscopic properties of aerosol particles collected at the ZOTTO site. As shown in Fig. 3,
both supermicrometer and submicrometer particles started to uptake substantial amount of
water at ~70% RH; nevertheless, efflorescence took place at different RH, with ERH being
~35% RH for submicrometer particles and ~50% RH for supermicrometer particles (Mikhailov
et al., 2015). It was suggested that the observed difference in ERH could be explained by the
difference in organic contents in submicrometer and supermicrometer particles (Mikhailov et
al., 2015): submicrometer particles contained larger fractions of organic materials,
consequently leading to the reduction of ERH.

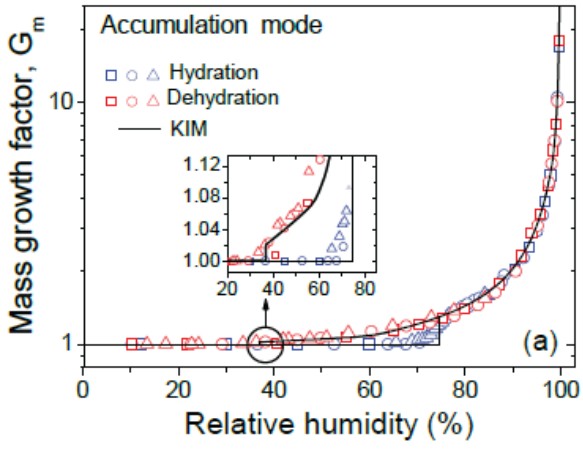

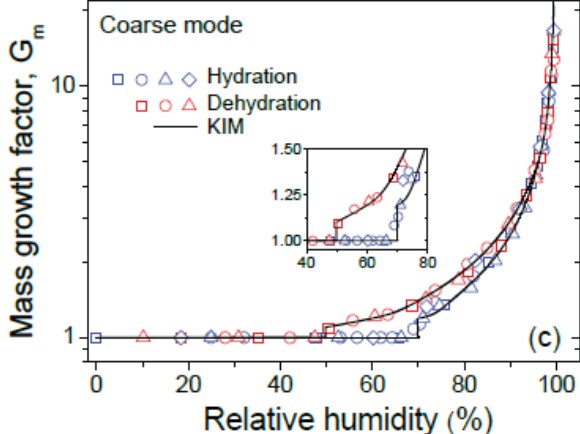




**Figure 3.** Mass growth factors of particles collected at the ZOTTO site in Serbia in June 2013:
(upper panel) accumulation mode; (lower panel) coarse mode. The solid curves represents
simulations using the $\kappa_\mathrm{m}$-interaction model (KIM). Reprinted with permission by Mikhailov et
al. (2015). Copyright 2015 Copernicus Publications.

The katharometer-based technique can be used to determine particle water content for
unsaturated and supersaturated samples, independent of particle size and morphology (Lee and
Chang, 2002; Mikhailov et al., 2013). It has also been successfully used as an offline method
to investigate hygroscopic properties of ambient aerosol particles (Mikhailov et al., 2013;
Mikhailov et al., 2015). It remains to be tested whether this technique is sensitive enough to
investigate water adsorption of a few monolayers or less.
**3.1.3 Knudsen cell reactor**
Knudsen cell reactors are low-pressure reactors widely used to investigate heterogeneous
uptake of trace gases (Al-Abadleh and Grassian, 2000; Karagulian and Rossi, 2005; Karagulian
et al., 2006; Wagner et al., 2008; Liu et al., 2009; Zhou et al., 2012). This technique was also
employed in several studies to explore water adsorption by particles with atmospheric
relevance (Rogaski et al., 1997; Seisel et al., 2004; Seisel et al., 2005). For example, the initial
uptake coefficient was reported to be 0.042±0.007 for uptake of water vapor by Saharan dust
at 298 K (Seisel et al., 2004). Another study (Rogaski et al., 1997) found that pretreatment with
$SO_2$, $HNO_3$ and $H_2SO_4$ could significantly increase water uptake by amorphous carbon.
Knudsen cell reactors are normally operated in the molecular flow regime, and thus water vapor
pressure used in these experiments is extremely low. As a result, although these measurements
can provide mechanistic insights into the interaction of water vapor with particles at the
molecular level, limited information on aerosol hygroscopicity under atmospheric conditions
can be provided.



### 3.2 Measurement of sample mass

Aerosol hygroscopicity can be quantitatively determined by measuring the mass of particles as a function of RH under isotherm conditions. This can be achieved by several types of experimental techniques, as introduced below.

### 3.2.1 Analytical balance

In a simple manner, the change in particle mass due to water uptake can be measured using an analytical balance under well controlled conditions (Hänel, 1976; McInnes et al., 1996; Hitzenberger et al., 1997; Diehl et al., 2001). For example, Diehl et al. (2001) investigated hygroscopic properties of ten pollen species at room temperature, using an analytical balance housed in a humidification chamber. The mass of pollen samples were measured at 0, (73±4) and (95±2)% RH. The average ratios of the mass of adsorbed water to dry mass increased from around 0.1 at 73% RH to ~3 at 95% RH (Diehl et al., 2001), suggesting that pollen samples can adsorb substantial amount of water at elevated RH.

Analytical balance was also employed to investigate hygroscopic properties of ambient aerosol particles. McInnes et al. (1996) employed an analytical balance to explore the hygroscopic properties of submicrometer marine aerosol particles collected on filters, and found that liquid water accounted for up to 9% of the dry particle mass at 35% RH and up to 29% of the dry particle mass at 47% RH. In another study (Hitzenberger et al., 1997), size-segregated aerosol particles were collected on aluminum foils using a nine-stage cascade impactor in downtown Vienna, and their hygroscopic properties were examined using an analytic balance. Aerosol hygroscopicity was found to be strongly size dependent (Hitzenberger et al., 1997), and the mass ratios of particles at 90% RH to that at dry condition were found to be 2.35-2.6 for particles in the accumulation mode and 1.16-1.33 for those in the coarse mode.



### 3.2.2 Thermogravimetric analysis

Similar to humidity-controlled analytical balance, thermogravimetric analysers (TGA) can directly measure the mass change of particle samples at different temperature to investigate aerosol hygroscopicity. Commercial TGA instruments are typically integrated with automated systems for humidity generation and control. They can control temperature and RH very precisely, and are very sensitive in mass measurement (typically down to 1 μg or even better).

Thermogravimetric analysers, sometimes also called vapor sorption analysers (VSA), have been employed by several groups to investigate hygroscopic properties of atmospherically relevant particles. For example, water uptake by $CaCO_3$ and Arizona test dust was measured at room temperature using a Mettler-Toledo TGA with an accuracy of 1 μg in mass measurement (Gustafsson et al., 2005), and about 4 monolayers of adsorbed water were formed at 80% RH for both mineral dust samples. A similar instrument was utilized to determine the DRH of dicarboxylic acids and their sodium salts at different temperatures (Beyer et al., 2014; Schroeder and Beyer, 2016), and the DRH was found to decrease with temperature for malonic acid, from 80.2% at 277 K to 69.5% at 303 K (Beyer et al., 2014). This method was also used to probe water adsorption by different soot particles (Popovitcheva et al., 2001; Popovicheva et al., 2008a; Popovicheva et al., 2008b), although no details of the instrument used were provided. It is worth noting that TGA and/or VSA have been widely used to investigate hygroscopic properties of pharmaceutical materials. For example, at room temperature anhydrous theophylline was observed to transform to hydrate at 62% RH, and its DRH was determined to be 99% (Chen et al., 2010).





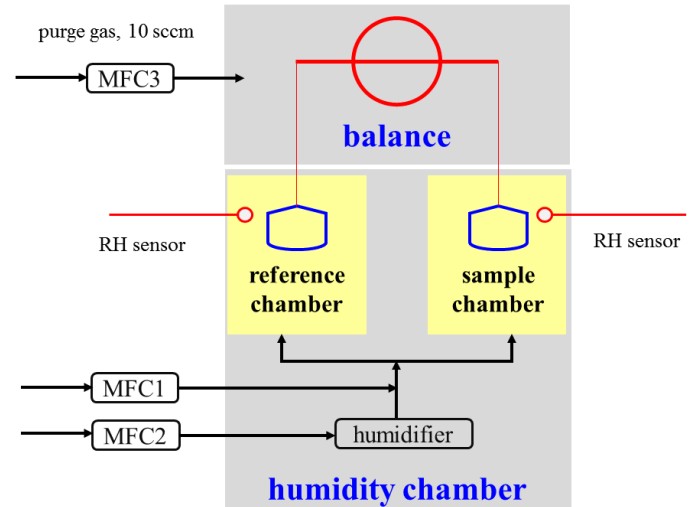

**Figure 4.** Schematic diagram of a vapor sorption analyser (Q5000SA, TA Instruments, New

Castle, DE, USA). Three mass flow controllers were used (MFC1: the dry flow; MFC2: the

humidified flow; MFC3: the dry flow to purge the balance). Reprint with permission by Gu et

al. (2017b). Copyright 2017 Copernicus Publications.

Very recently, Tang and coworkers systematically evaluated the performance of a vapor

sorption analyser to investigate hygroscopic properties of particles of atmospheric relevance

(Gu et al., 2017b). The instrument, with its schematic diagram shown in Fig. 4, has two sample

crucibles housed in a temperature- and humidity-regulated chamber, and one crucible is empty

so that the background is simultaneously measured and subtracted. DRH values of six

compounds, including $(NH_4)_2SO_4$ and NaCl, were determined at different temperatures (5-30

ᵒC) and found to agree well with literature values. In addition, the mass change as a function

of RH (up to 90%), relative to that at 0% RH, was also found to agree well with those calculated

using the E-AIM model (Clegg et al., 1998) for $(NH_4)_2SO_4$ and NaCl at 5 and 25 ᵒC. Therefore,

it can be concluded that the vapor sorption analyser is a reliable technique to study hygroscopic

properties of atmospherically relevant particles.



The vapor sorption analyzer was used to examine hygroscopicity of CaSO$_4$·2H$_2$O at 25 °C
(Gu et al., 2017b), and the results are displayed in Fig. 5. The hygroscopicity of CaSO$_4$·2H$_2$O
was found to be very low, and the sample mass was only increased by <0.5% when RH was
increased from 0 to 95%. This instrument was very sensitive to the change in sample mass due
to water uptake; for example, as shown in Fig. 5b, a relative mass change of <0.025% within 6
h could be accurately determined. This instrument was further employed to investigate
hygroscopic properties of perchlorates (Gu et al., 2017a; Jia et al., 2018), Ca- and Mg-
containing salts (Guo et al., 2019), and primary biological particles (Tang et al., 2019), which
play significant roles in the environments of the earth and the Mars. To our knowledge, the
VSA technique has not yet been used to explore hygroscopic properties of ambient aerosol
particles.

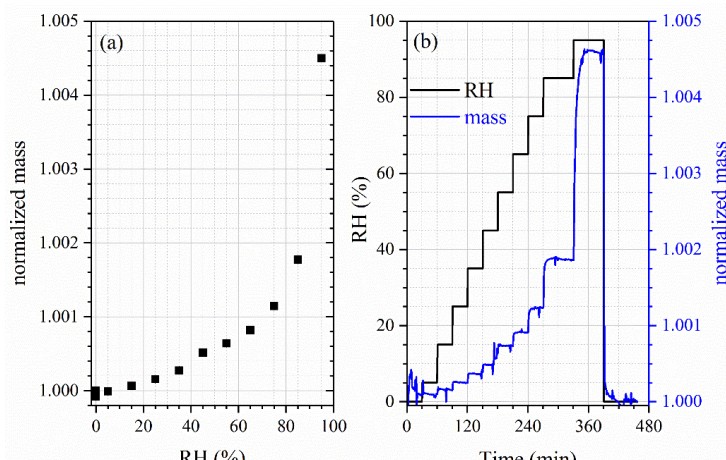


**Figure 5.** Sample mass of CaSO$_4$·2H$_2$O (relative to that of 0% RH) as a function of RH at 25
°C, measured using a vapor sorption analyzer. (a) Change of sample mass with RH up to 95%;
(b) change of sample mass and RH with experimental time. Reprint with permission by Gu et
al. (2017b). Copyright 2017 Copernicus Publications.





### 3.2.3 Quartz crystal microbalance

It was proposed in 1959 (Sauerbrey, 1959) that a film attached to the electrodes of a piezoelectric quartz resonator would cause a decrease in the resonance frequency, given by Eq. (1):

$$\Delta f = -C_f \cdot \Delta m \quad (1)$$

where $\Delta f$ is the change in resonance frequency, $\Delta m$ is the mass of the film, and $C_f$ is a constant specific for the quartz resonator and can be experimentally calibrated. Eq. (1), known as the Sauerbrey equation, forms the basis for using the piezoelectric quartz resonator as a microbalance, which is usually called quartz crystal microbalance (QCM). QCM is a highly sensitive technique for particle mass measurement, and could be extended to investigate aerosol hygroscopicity. In a typical experiment, a particle film is first coupled to the quartz crystal, and RH is then varied with the resonance frequency being simultaneously recorded. According to Eq. (1), change in the mass of the particle film, due to change in RH, is proportional to the change in resonance frequency. Hygroscopicity measurements only need the information of relative mass change (relative to that under dry conditions), and as a result, knowledge of $C_f$ is not required. QCM has a very high sensitivity in mass measurement, and it has been reported that the change in mass on the order of a few percent of a monolayer can be reliably determined (Tsionsky and Gileadi, 1994).

A QCM was used to measure the DRH of a number of inorganic and organic salts, including NaCl, $(NH_4)_2SO_4$, $CH_3COONa$ and $CH_3COOK$ (Arenas et al., 2012), and the measured values agreed very well with those reported in previous work. Several studies (Thomas et al., 1999; Demou et al., 2003; Asad et al., 2004a; Liu et al., 2016) have utilized QCM to explore hygroscopic properties of organic compounds of atmospheric interest. For example, Demou et al. (2003) quantitatively determined the amount of water taken up by dodecane, 1-octanol, octanoic acid, 1,5-pentanediol, 1,8-octanediol and malonic acid at room

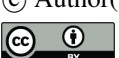


temperature. The DRH was measured to be ~72% for malonic acid and ~95% for 1,8-octanediol,
and in general compounds with higher oxidation state showed higher hygroscopicity (Demou
et al., 2003). Another study (Asad et al., 2004a) found that exposure to $O_3$ would substantially
increase the hygroscopicity of oleic acid. Using a QCM, Zuberi et al. (2005) explored the effect
of heterogeneous reactions on hygroscopic properties of soot particles. As shown in Fig. 6,
while water adsorption was very limited for fresh soot particles, hygroscopicity of soot particles
was significantly increased after heterogeneous reactions with OH/$O_3$ and $HNO_3$ (Zuberi et al.,

2005).

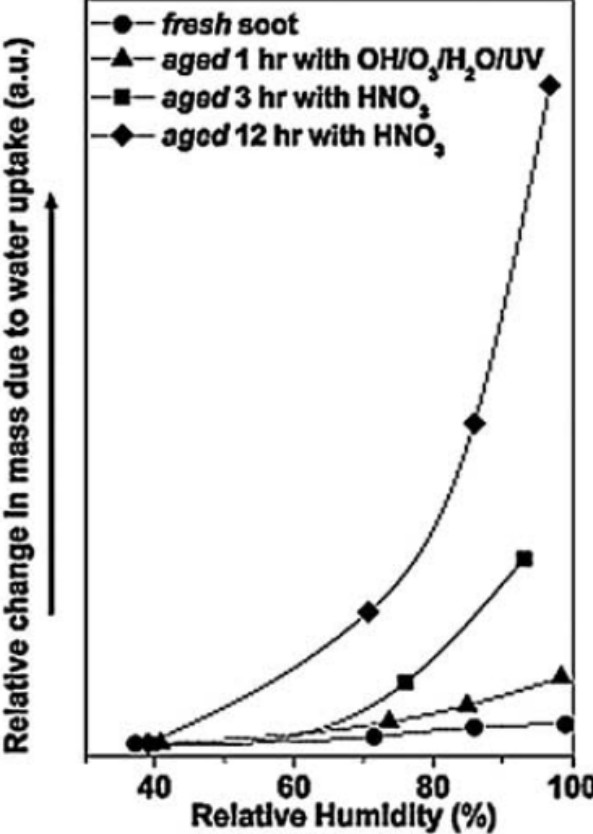




**Figure 6.** Water uptake (quantified as the ratio of mass of water taken up to that of dry particle
mass) of fresh and aged soot particles. Reprinted with permission by Zuberi et al. (2005).
Copyright 2005 John Wiley & Son, Inc.

QCM has also been applied to study hygroscopic properties of mineral dust particles,

including oxides (Schuttlefield et al., 2007a), clay minerals (Schuttlefield et al., 2007b;
Yeşilbaş and Boily, 2016) and authentic dust samples (Navea et al., 2010; Yeşilbaş and Boily,
2016). For example, Yeşilbaş and Boily (2016) measured the amount of water taken up by 21
different types of mineral particles up to 70% RH at 25 °C, and found that particle size played
a critical role in water adsorption by these minerals. At 70% RH, submicrometer-sized particles
could adsorb up to ~5 monolayers of water, while the amount of water adsorbed by micrometer-
sized particles can reach several thousand monolayers (Yeşilbaş and Boily, 2016). Another
study (Hatch et al., 2008) suggested that ~3 monolayers of adsorbed water was formed on
$CaCO_3$ particles at 78% RH, and internal mixing with humic and fulvic acids could
substantially increase the hygroscopicity of $CaCO_3$.

It should be pointed out (as often not fully considered) that a few assumptions are required

for the Sauerbrey equation to be valid (Rodahl and Kasemo, 1996), including: (i) the film
deposited on the quartz crystal is rigid, i.e. internal friction is negligible; (ii) the film is perfectly
coupled to the quartz crystal, i.e. there is no slip between the film and the crystal. The Sauerbrey
equation may not hold if these conditions are not fulfilled, and the stiffness of the particle film
would significantly affect the quartz resonator response (Dybwad, 1985; Pomorska et al., 2010;
Vittorias et al., 2010; Arenas et al., 2012). Rodal and Kasemo (1996) suggested that the
Sauerbrey equation can offer reliable mass change measurement only if the film is thin enough
and does not slide on the QCM electrode. In addition, as supersaturated films formed on the



quartz crystal are unstable, QCM may not be able to explore hygroscopic properties of
supersaturated samples.

Piezoelectric bulk wave resonators, which work in a way similar to the QCM, have been

used for monitoring aerosol mass concentrations (Thomas et al., 2016; Wasisto et al., 2016).
When particles are deposited onto the resonator surface, the resonance frequency will be
linearly reduced with the particle mass. Very recently a new method based on piezoelectric
bulk wave resonators was developed to investigate aerosol hygroscopicity (Zielinski et al.,
2018). Aerosol particles were first collected on the resonator surface and then exposed to
changing RH. Measured DRH and ERH values were found to agree with literature for NaCl
and $(NH_4)_2SO_4$; in addition, good consistency between experimentally measured and E-AIM
predicted hygroscopic growth curves was found for NaCl, $(NH_4)_2SO_4$ and NaCl/malonic acid
mixture (Zielinski et al., 2018). Therefore, this technique appears to be a very promising
method for aerosol hygroscopicity measurements.
**3.2.4 Beta gauge and TEOM**

In addition to the gravimetric method, the beta gauge method is widely used to measure

aerosol mass concentrations in a semi-continuous way (Courtney et al., 1982; Chow, 1995;
McMurry, 2000; Solomon and Sioutas, 2008; Kulkarni et al., 2011). A beta gauge measures
the attenuation of beta particles emitted from a radioactive source through a particle-loaded
filter, and if properly calibrated, attenuation of beta particles through the filter can be used to
quantify the mass of particles loaded on the filter (McMurry, 2000). The mass of aerosol
particles, after being collected on a filter, was measured at different RH in a closed chamber
using a beta gauge to determine the aerosol liquid water content (Speer et al., 1997). Laboratory
evaluation showed that the liquid water content of $(NH_4)_2SO_4$ determined using this method
agreed well with those measured gravimetrically (Speer et al., 1997), and when compared to
humidification, a hysteresis was found during dehumidification for $(NH_4)_2SO_4$. The ability to



observe hysteresis is related to the use of hydrophobic substrate (for example, Teflon is usually
a good option) in particle sampling. In addition, the beta gauge method was preliminarily
employed to explore hygroscopic properties of submicrometer ambient aerosol particles (Speer
et al., 1997). Further tests with other compounds, in addition to $(NH_4)_2SO_4$, are required to
validate the robustness and reliability of this method.

Another widely-employed semi-continuous technique for aerosol mass measurement is

tapered-element oscillating microbalance (TEOM) (Patashnick and Rupprecht, 1991; Chow et
al., 2008; Solomon and Sioutas, 2008; Kulkarni et al., 2011). In a typical TEOM instrument,
the wide end of a tapered hollow tube is mounted on a base plate, and its narrow end is coupled
to a filter used to collected aerosol particles (Kulkarni et al., 2011). The oscillation frequency
of the tapered hollow tube depends on the mass of particles collected on the filter and can be
used to measure particle mass if properly calibrated (Kulkarni et al., 2011). Rogers et al. (1998)
explored the possibility of using TEOM to measure aerosol liquid water content. Increase in
particle mass was observed when a humid particle-free air flow was passed through a particle-
loaded filter in the TEOM, and the particle mass started to decrease after a dry particle-free air
was introduced (Rogers et al., 1998). This suggested that TEOM had the potential to examine
hygroscopic properties of aerosol particles, though further experimental evaluation is needed
to assess its performance.
**3.2.5 Discussion**

All the techniques discussed in Section 3.2 determine particle water content through direct

measurement of sample mass or properties that are related to the sample mass, and hence there
is no requirement on particle shape. Some of these techniques, such as thermogravimetric
analysis (Gustafsson et al., 2005) and quartz crystal microbalance (Schuttlefield et al., 2007a;
Yeşilbaş and Boily, 2016), are sensitive enough to investigate water adsorption down to one or
a few monolayers, while other techniques, such as the analytic balance, may not be sensitive



enough for this application. If particles are supported on proper substrates (such as hydrophobic
films), these techniques can be used to investigate hygroscopic properties of supersaturated
samples, as demonstrated for the beta gauge method (Speer et al., 1997) and the piezoelectric
bulk wave resonators (Zielinski et al., 2018). Nevertheless, supersaturated solutions formed in
majority of these applications may not be stable enough for hygroscopic growth measurements,
and as a result measurements have been rarely reported for supersaturated samples. In principle
these techniques can all be used offline to investigate ambient aerosol particles if samples with
enough mass can be collected. Analytical balance (McInnes et al., 1996; Hitzenberger et al.,
1997) and the beta gauge method (Speer et al., 1997) have been used to explore hygroscopic
properties of ambient aerosols; to our knowledge, application of thermogravimetric analysis,
quartz crystal microbalance, TOEM and piezoelectric bulk wave resonators to ambient samples
is yet to be demonstrated.
**3.3 Microscopic techniques**

Deliquescence and efflorescence can be monitored using a number of microscopic

methods, as discussed in this section. Furthermore, change in particle size at different RH, as
measured microscopically, can be used to determine hygroscopic growth factors.
**3.3.1 Optical microscopy**

Optical microscopy was employed to investigate phase transition of atmospheric particles

as early as in 1950s (Twomey, 1953; Twomey, 1954). In these two studies (Twomey, 1953;
Twomey, 1954), a large number of aerosol particles collected in Sydney were found to
deliquesce at 71-75% RH, implying that they consisted mainly of sea salt. Since then, optical
microscopy has been widely used to study hygroscopic properties of atmospherically relevant
particles, and herein we only introduce representative studies conducted in the last two decades.

Bertram and co-workers (Parsons et al., 2004a; Parsons et al., 2004b; Parsons et al., 2006)

developed a flow cell-optical microscope apparatus to investigate phase transitions of



individual particles deposited on glass slides coated with hydrophobic films. As show in Fig.
7, the glass slide was placed in a flow cell mounted on a cooling stage for temperature
regulation. A dry nitrogen flow was mixed with a humidified nitrogen flow and then delivered
into the flow cell through the inlet, and the two flows were regulated using two mass flow
controllers to adjust water vapor pressure (and thus RH) in the flow cell. Phase transitions of
particles deposited on the glass slide were monitored using a microscope, and particle images
were recorded using a CCD camera.

(a)

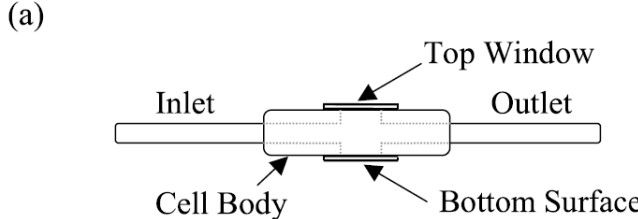

(b)

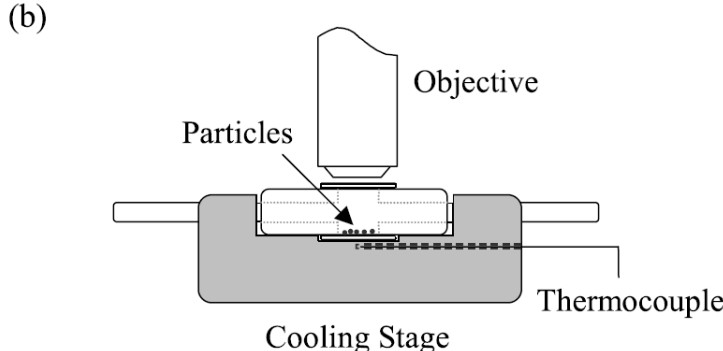


**Figure 7.** Schematic diagram of the flow cell-optical microscope apparatus developed by
Bertram and co-workers to investigate particle phase transitions: (a) side view of the flow cell;
(b) side view of the entire apparatus. Particles were deposited on a glass slide placed on the
bottom of the flow cell, which was mounted on a cooling stage. Objective: objective lens of





the microscope. Reprint with permission by Parsons et al. (2004b). Copyright 2004 John Wiley
& Sons, Inc.

The performance of this apparatus was evaluated by measuring DRH of $(NH_4)_2SO_4$
particles from ~260 to 300 K (Parsons et al., 2004b), and the measured DRH agreed well with
those reported in literature. This setup was then used to investigate the deliquescence of
malonic, succinic, glutaric and adipic acid particles from 243 to 293 K  (Parsons et al., 2004b)
and deliquescence and crystallization of $(NH_4)_2SO_4$ and NaCl particles internally mixed with
organic compounds (Pant et al., 2004; Parsons et al., 2004a). It was found that if $(NH_4)_2SO_4$ or
NaCl particles contained substantial amounts of organic materials, their DRH would be
significantly reduced and these particles were more likely to be aqueous in the troposphere
(Pant et al., 2004). A similar instrument was employed to investigate deliquescence and
efflorescence of $HIO_3$ and $I_2O_5$ particles (Kumar et al., 2010), and the DRH at 293 K were
reported to be 81% for $HIO_3$ and 85% for $I_2O_5$.
As illustrated by Fig. 8a, besides deliquescence and efflorescence, atmospheric aerosols
can also undergo liquid-liquid phase separation (LLPS), leading to coexistence of two liquid
phases (Bertram et al., 2011; You et al., 2012; You et al., 2014; Freedman, 2017). LLPS can
impact the direct and indirect radiative forcing of atmospheric aerosol particles as well as their
heterogeneous reactivity, and therefore has received increasing attention in the last several
years (You et al., 2012; Freedman, 2017). Optical microscopy has played an important role in
understanding LLPS of atmospherically relevant particles (Bertram et al., 2011; You et al.,
2012; You et al., 2014). Fig. 8b shows optical microscopic images of an internally mixed
particle during an experiment in which RH was decreased while temperature was kept at ~291
K (Bertram et al., 2011), and the particle contained $(NH_4)_2SO_4$ and 1,2,6-trihydroxyhexane
with a mass ratio of 1:2.1. As shown in Fig. 8b, at high RH the particle existed as an aqueous



droplet, and LLPS happened when RH was decreased, leading to the formation of two liquid
phases; efflorescence took place with further decrease in RH, leading to the formation of a solid
$(NH_4)_2SO_4$ core coated with an organic liquid layer.

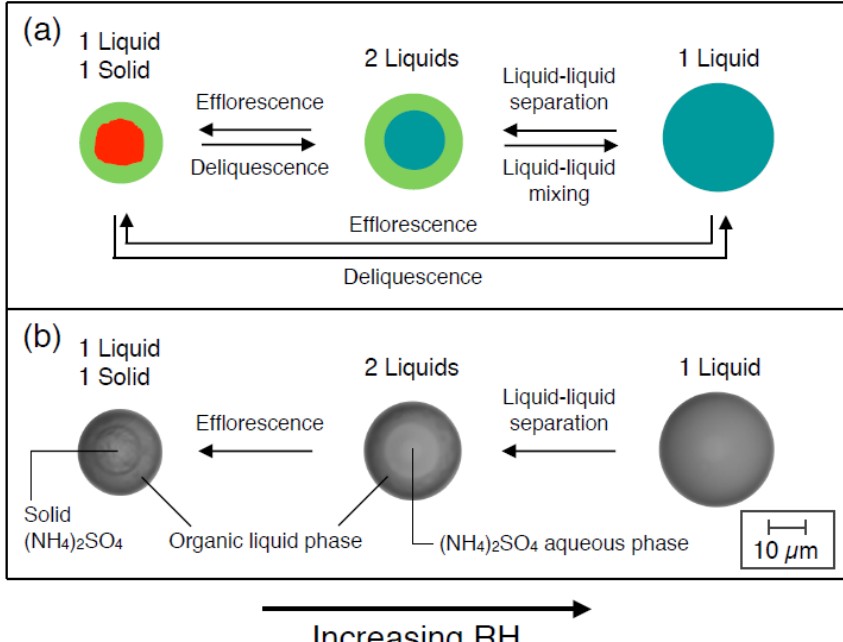


**Figure 8.** (a) Some of the phase transitions which may occur for internally mixed atmospheric

particles consisting of $(NH_4)_2SO_4$ and organic materials. Aqua represents an aqueous phase,
green represents a liquid phase of organic material, and red presents a solid phase of $(NH_4)_2SO_4$.
(b) Optical microscopic images of a particle which contained $(NH4)_2SO_4$ and 1,2,6-
trihydroxyhexane with a mass ratio of 1:2.1, during an experiment in which temperature was
kept at around 291 K while RH was decreased. Reprint with permission by Bertram et al. (2011).
Copyright 2011 Copernicus Publications.

In addition to identification of phase transitions, analysis of optical microscopic images

recorded can also be used to determine particle size change and as a result hygroscopic growth
factors (Ahn et al., 2010; Eom et al., 2014; Gupta et al., 2015). For instance, Ahn et al. (2010)



employed an optical microscope to investigate hygroscopic properties of NaCl, KCl,
$(NH_4)_2SO_4$ and $Na_2SO_4$ particles collected on TEM grids, and found that their measured
hygroscopic growth factors agreed well with those reported in literature for all the four types
of particles examined. A following study (Eom et al., 2014) compared the influence of six types
of supporting substrates (including TEM grid, Parafilm-M, aluminum foil, Ag foil, silicon
wafer and cover glass) on hygroscopicity measurements using optical microscopy, and
concluded that TEM grids were the most suitable substrate for this application. Optical
microscopy was also used to study hygroscopic properties of $MgCl_2$ and $NaCl$-$MgCl_2$ mixed
particles (Gupta et al., 2015), and hygroscopic properties (including DRH and growth factors)
of these particles were found to differ significantly from NaCl. Since $MgCl_2$ is an important
component in sea salt aerosol, this work can have significant implications for hygroscopicity
and thus climatic impacts of sea salt aerosol (Zieger et al., 2017).

Optical microscopy can be (and has been widely) coupled to suitable spectroscopic

techniques such as FTIR (Liu et al., 2008a), Raman spectroscopy (Liu et al., 2008c) and
fluorescence (Montgomery et al., 2015), and if so chemical information can be simultaneously
provided.
**3.3.2 Electron microscopy**

Electron microscopy has been widely used in laboratory and field studies to examine

composition, mixing state and morphology of atmospheric particles, as summarized by a few
excellent review articles (Prather et al., 2008; Posfai and Buseck, 2010; Li et al., 2015; Ault
and Axson, 2017). Herein we discuss exemplary studies to illustrate how electron microscopy
can help improve our knowledge of aerosol hygroscopicity. This section is further divided to
two parts, i.e. scanning electron microscopy (SEM) and transmission electron microscopy
(TEM).



### 3.3.2.1 SEM

Ebert et al. (2002) developed an environmental scanning electron microscope (ESEM) technique to explore hygroscopic properties of individual particles, and the instrument they used had a spatial resolution of 8-15 nm. Changes in particle morphology could be used to identify phase transitions (deliquescence and efflorescence), and growth factors could be derived from observed change in particles size at different RH. Their measured DRH and hygroscopic growth factors (Ebert et al., 2002) were in good agreement with results reported by previous literature for NaCl, $(NH_4)_2SO_4$, $Na_2SO_4$ and $NH_4NO_3$. However, ERH could not be accurately determined due to the influence of the substrate onto which particles under investigation were deposited (Ebert et al., 2002).

ESEM, coupled to energy disperse X-ray analysis (EDX), was employed to investigate hygroscopic properties of a wide range of atmospheric particles, including $(NH_4)_2SO_4$ (Matsumura and Hayashi, 2007), sea spray (Hoffman et al., 2004), aerosol particles collected in nickel refineries (Inerle-Hof et al., 2007), agricultural aerosol (Hiranuma et al., 2008), pollen (Pope, 2010; Griffiths et al., 2012) and protein (Gomery et al., 2013). For example, Hoffman et al. (2004) found that both $NaNO_3$ and $NaNO_3/NaCl$ particles existed as amorphous solids even at very low RH and exhibit continuous hygroscopic growth, instead of having clear DRH; furthermore, EDX analysis showed that Cl was enriched in the core of dried $NaCl/NaNO_3$ particles (Hoffman et al., 2004), implying that during dehumidification NaCl started to crystalline first because of its lower solubility. This finding may have important implications for chemical and radiative properties of marine aerosol particles (Quinn et al., 2015). In another study (Pope, 2010), ESEM observations revealed that birch pollen gains swelled internally but did not take up water on the surface significantly even at 93% RH; however, liquid water could be observed on the particle surface when RH was >95%. Hiranuma et al. (2008) found that most of aerosol particles collected at a cattle feedlot in the Texas did not take up significant



amount of water at 96% RH, though a small fraction of coarse particles became deliquesced at
~75% RH and their sizes were doubled at 96% RH compared to their original sizes.

SEM/EDX was utilized by Krueger et al. (2003) to monitor changes in phase, morphology

and composition of individual mineral dust particles after heterogeneous reaction with gaseous
$HNO_3$. For the first time, laboratory work showed that solid mineral dust particles could be
transformed to aqueous droplets due to heterogeneous reactions (Krueger et al., 2003). As
displayed in Fig. 9, solid $CaCO_3$ particles were converted to spherical droplets as
heterogeneous reaction with gaseous $HNO_3$ proceeded (Krueger et al., 2003), and this was
caused by the formation of $Ca(NO_3)_2$ which had very low DRH (Al-Abadleh et al., 2003; Kelly
and Wexler, 2005). A following study (Krueger et al., 2004) examined heterogeneous reactions
of $HNO_3$ with mineral dust samples collected from four different regions, using SEM/EDX. It
was suggested that calcite and dolomite particles exhibited large reactivity towards $HNO_3$ and
could be transformed to aqueous droplets, while no morphological change was observed for
gypsum, aluminum silicate clay and quartz particles after exposure to $HNO_3$ (Krueger et al.,

2004).

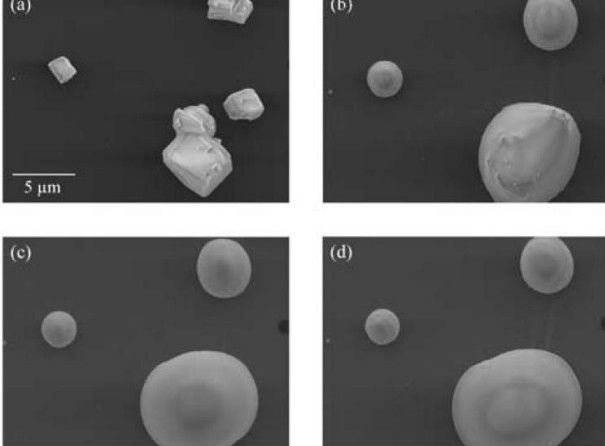




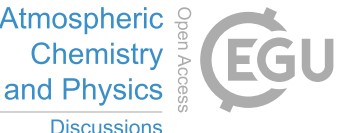

**Figure 9.** SEM images of $CaCO_3$ particles before and after exposure to 26 ppbv gaseous $HNO_3$
at ~41% RH. (a): Before exposure; (b) exposure for 1 h; (c) exposure for 2 h; (d) exposure for
4 h. Reprint with permission by Krueger et al. (2003). Copyright 2003 John Wiley & Sons, Inc.

The new laboratory discovery by Krueger et al. (2003) has been supported by a number of

field measurements (Li et al., 2015; Tang et al., 2016a), and in some of which SEM was also
utilized. For example, Laskin et al. (Laskin et al., 2005) provided the first evidence
demonstrating that in the ambient air solid nonspherical $CaCO_3$ particles could be transformed
to aqueous droplets which contained $Ca(NO_3)_2$ formed in heterogeneous reaction with nitrogen
oxides. ESEM was also applied to examine mineral dust particles collected in Beijing (Matsuki
et al., 2005) and southwestern Japan (Shi et al., 2008), and both studies found that  some Ca-
containing particles existed in aqueous state even at RH as low as 15% because heterogeneous
reactions with nitrogen oxides converted $CaCO_3$ to $Ca(NO_3)_2$. Similarly, it was shown by
SEM/EDX measurements (Tobo et al., 2010; Tobo et al., 2012) that Ca-containing mineral
dust particles in remote marine troposphere were transformed to aqueous droplets, because
$CaCl_2$ was formed in heterogeneous reaction of $CaCO_3$ with HCl.
**3.3.2.2 TEM**

Compared to SEM, transmission electron microscopy (TEM) has better spatial resolution

and can resolve features down to one nanometer or even smaller. TEM and AFM (atomic force
microscopy) were employed by Buseck and colleagues (Posfai et al., 1998) to examine ambient
particles collected on TEM grids under vacuum and ambient conditions. It was found that
particle volumes were up to four times larger under ambient conditions, compared to vacuum
conditions. Several years later Buseck and co-workers (Wise et al., 2005) developed an
environmental transmission electron microscope (ETEM) which enabled individual particles
to be characterized under environmental conditions. The performance of this instrument was





validated by measuring DRH and ERH of NaBr, CsCl, NaCl, $(NH_4)_2SO_4$ and KBr particles in
the size range of 0.1-1 μm, and good agreement was found between their measured values and
these reported by previous work for all the five compounds investigated (Wise et al., 2005).

The ETEM technique was further employed to investigate hygroscopic properties of a

wide range of atmospheric particles, including NaCl-containing particles (Semeniuk et al.,
2007b; Wise et al., 2007), biomass burning particles (Semeniuk et al., 2007a) and potassium
salts (Freney et al., 2009). The DRH of NaCl particles internally mixed with insoluble materials
was determined to be ~76% (equal to that for pure NaCl), while internal mixing with other
soluble compounds (e.g., $NaNO_3$) would reduce the DRH (Wise et al., 2007). DRH and ERH
were reported to be 85 and 56% for KCl and 96 and 60% for $K_2SO_4$, while $KNO_3$ displayed
continuous hygroscopic growth (Freney et al., 2009); in addition, deliquescence and
efflorescence of internally mixed $KCl/KNO_3$ and $KCl/K_2SO_4$ were also examined (Freney et
al., 2009). In another study (Adachi et al., 2011), aerosol particles, mainly being sulfate
internally mixed with weakly hygroscopic organic materials, were collected at Mexico City
and their hygroscopic properties were investigated using ETEM. It was found that only the
sulfate part was deliquesced at elevated RH, while the entire particles containing deliquesced
sulfate did not necessarily became spherical. It was further suggested that the actual light
scattering ability was 50% larger than that estimated by Mie theory which assumes particle
sphericity (Adachi et al., 2011).

Recently cryogenic TEM has been deployed to explore morphology, hygroscopic

properties and chemical composition of atmospheric particles (Veghte et al., 2014; Patterson
et al., 2016). For example, it was observed that most nascent sea spray aerosol particles were
homogeneous aqueous droplets, and upon exposure to low RH they would be quickly
reorganized and undergo phase separation (Patterson et al., 2016).



### 3.3.3 Atomic force microscopy


Atomic force microscopy (AFM) is a widely used technique in surface chemistry and
surface science. Compared to other microscopic techniques (e.g., optical microscopy, FTIR
microscopy, TEM and SEM), AFM has several unique advantages. It does not require vacuum
condition, and thus can be operated under environmental conditions; in addition, it has a high
spatial resolution down to the nanometer level, and offers three-dimensional imaging (Morris
et al., 2016).
In the past two decades, AFM has been gradually utilized in atmospheric chemistry to
observe three-dimensional morphology of aerosol particles, and its application in atmospheric
chemistry started with observation of surfaces of single crystals with atmospheric relevance.
For example, AFM was employed to study the (100) cleavage surface of NaCl during exposure
to water vapor (Dai et al., 1997). A uniform layer of water was formed on the surface and
surface steps started to evolve slowly at ~35% RH; when RH increased to ~73%
(approximately the DRH of NaCl), the step structure disappeared abruptly due to deliquescence
of the surface (Dai et al., 1997). This pioneering work demonstrated that AFM had the potential
to be used to determine DRH of hygroscopic salts, in addition to providing rich information of
surface structure change during exposure to water vapor. AFM was later used to observe
MgO(100) and CaCO$_3$(1014) surface during exposure to water vapor and gaseous nitric acid
(Krueger et al., 2005). Instabilities of oscillations in AFM images were observed, indicating
that deliquescence of nitrate salts, which were formed in to heterogeneous reaction with nitric
acid, occurred at elevated RH (Krueger et al., 2005).
To our knowledge, AFM was successfully used in 1995 to characterize aerosol particles
collected using a low-pressure impactor (Friedbacher et al., 1995). Three years later, Posfai et
al. (1998) used AFM to examine individual particles collected above the North Atlantic Ocean
at different RH. The particle volume was observed to be four times larger under ambient



conditions (measured by AFM) compared to that in the vacuum (measured by TEM) (Posfai et
al., 1998). Another study (Wittmaack and Strigl, 2005) used AFM to measure height-to-
diameter ratios of ambient particles, and concluded that some particles may exist in the
supersaturated metastable state at around 50% RH. Non-contact environmental AFM was used
to examine uptake of water vapor by NaCl nanoparticles at RH below DRH (Bruzewicz et al.,
2011). NaCl nanoparticles started to adsorb water at RH well below its DRH (75%), and a
liquid-like surface layer with thickness of 2-5 nm was formed at 70% RH, suggesting that
deliquescence of NaCl nanoparticles was much more complicated than an abrupt first-order
phase transition.
Very recently Tivanski and co-workers (Ghorai et al., 2014; Laskina et al., 2015b; Morris
et al., 2015; Morris et al., 2016) developed an AFM-based method to investigate hygroscopicity
of particles deposited on substrates, and systematically evaluated its performance by measuring
hygroscopic growth factors of NaCl, malonic acid and binary mixture of NaCl with malonic or
nonanoic acid. It was found that hygroscopic growth factors derived from 3D volume
equivalent diameters always agreed well with H-TDMA results; however, hygroscopic growth
factors derived from 2D area equivalent diameters showed significant deviation from H-TDMA
results for some types of particles (Morris et al., 2016). An example is displayed in Fig. 10,
suggesting that at 80% RH, the hygroscopic growth factor of NaCl particles derived from the
volume-equivalent diameter was equal to that determined using H-TDMA, significantly larger
than that derived from area-equivalent diameter. Such deviation was caused by anisotropic
growth of particles (Morris et al., 2016), and the extent of deviation depended on the particle
composition and their hydrate state at the time when they were collected on the substrate.





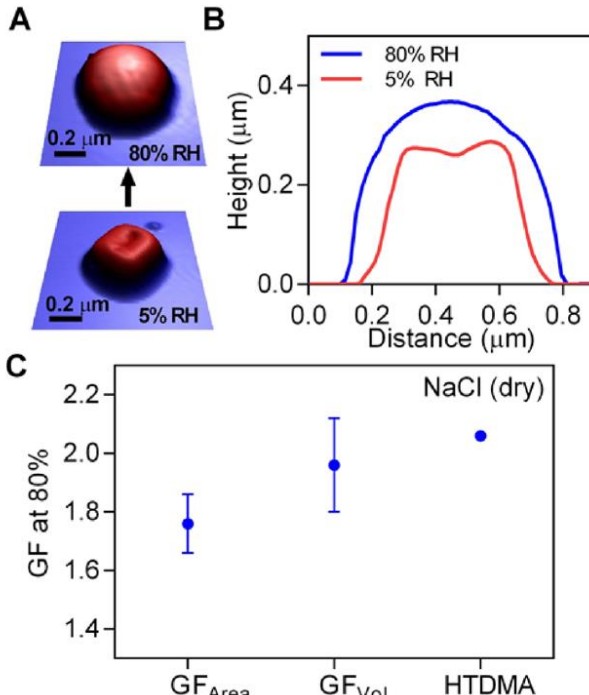


**Figure 10.** AFM measurements of hygroscopicity of NaCl particles. (A) 3D AFM images of a

NaCl particle at 5 and 80% RH; (B) Cross section of the particles at 5% (red) and 80% (blue)
RH; (C) Comparison of hygroscopic growth factors derived from changes in mobility diameter
(measured using H-TDMA), area equivalent diameter (measured using AFM) and volume
equivalent diameter (measured using AFM). Reprint with permission by Morris et al. (2016).
Copyright 2016 American Chemical Society.

In addition to hygroscopicity measurement, AFM were used in several studies to
characterize morphology, structure and other physicochemical properties of atmospheric
particles (Lehmpuhl et al., 1999; Freedman et al., 2010; Laskina et al., 2015a). For example,
AFM measurements found that organic and soot particles would shrink after interactions with
$O_3$ while inorganic particles remained unchanged (Lehmpuhl et al., 1999). Freedman et al.
(2010) employed AFM coupled to Raman microscopy to characterize atmospheric particles





under ambient conditions, and observed core-shell structure for some organic particles. A
recent study (Laskina et al., 2015a) characterized particles collected on substrates using AFM,
Raman microscopy and SEM, and suggested that microscopy techniques operated under
ambient conditions would offer the most relevant and robust information on particle size and
morphology. Conventional AFM offers no chemical information; however, it can be (and has
already been) coupled to spectroscopic techniques (such as FTIR) (Dazzi et al., 2012; Ault and
Axson, 2017; Dazzi and Prater, 2017), enabling detailed physical and chemical properties to
be provided with high spatial resolution. Very recently, the peak force infrared microscopy, a
type of scanning probe microscopy, was developed to investigate IR absorption and mechanical
properties of ambient aerosol particles (Wang et al., 2017b), and a spatial resolution of 10 nm
could be achieved.
**3.3.4 X-ray microscopy**
Scanning transmission X-ray microscopy (STXM) is a novel technique which can provide
spatial distribution of physical, chemical and morphological information of individual particles
(de Smit et al., 2008), and has been recently employed to investigate atmospheric particles
(Ault and Axson, 2017). For example, Ghorai and Tivanski (2010) developed a STXM-based
method to study hygroscopic growth of individual submicrometer particles, and proposed a
method to quantify the mass of water associated with individual particles at a given RH. DRH
and ERH values of NaCl, NaBr, and $NaNO_3$, determined using STXM (Ghorai and Tivanski,
2010), agreed very well with previous results, and mass hygroscopic growth factors were also
reported for these particles. In a following study (Ghorai et al., 2011), STXM was used to
investigate hygroscopic growth of individual malonic acid; in addition to measured mass
hygroscopic growth factors, near-edge X-ray absorption fine structure spectroscopy (NEXAFS)
acquired using STXM suggested that keto-enol tautomerism occurred for deliquesced malonic





acid particles (Ghorai et al., 2011). The keto-enol equilibrium constants were found to vary
with RH, with enol formation favored at high RH (Ghorai et al., 2011).

Hygroscopic growth of submicrometer $(NH_4)_2SO_4$, measured using STXM/NEXAFS

(Zelenay et al., 2011a), agreed well with previous studies; furthermore, analysis of STXM
images and NEXAFS spectra suggested that phase separation occurred for internally mixed
$(NH_4)_2SO_4$-adipic acid particles, and adipic acid was partially enclosed by $(NH_4)_2SO_4$ at high
RH (Zelenay et al., 2011a). An environmental chamber was constructed to be directly coupled
to a STXM instrument (Kelly et al., 2013), and this set-up was utilized to explore hygroscopic
properties of NaCl, NaBr, KCl, $(NH_4)_2SO_4$, levoglucosan and fructose (Piens et al., 2016).
Measured mass hygroscopic growth factors were compared with those predicted by a
thermodynamic model (AIOMFAC) (Zuend et al., 2011), and good agreement between
measurement and prediction was found for all the compounds investigated (Piens et al., 2016).
In another study, Zelenay et al. (2010b) utilized STXM/NEXAFS to investigate hygroscopic
properties of submicrometer tannic acid and Suwannee River Fulvic acid used as proxies for
humic-like substance found in atmospheric aerosol. Both compounds exhibited continuous
water uptake, and at 90% RH around one water molecule was associated with each oxygen
atoms contained by tannic acid while approximately two water molecules were associated with
each oxygen atoms contained by Suwannee River Fulvic acid (Zelenay et al., 2011b).





**Figure 11.** Hygroscopicity, mass growth factors at 80% RH ($g_m$), single hygroscopicity parameters ($\kappa_{eqiv}$), inorganic atomic fractions, STXM images (acquired at 4 and 90% RH) and mixing state for 15 aerosol particles examined. Reprint with permission by Piens et al. (2016). Copyright 2016 American Chemical Society.

STXM/NEXAFS has already been applied to explore hygroscopicity of ambient particles. For example, Pöhlker et al. (2014) collected aerosol particles from the Amazonian forest during periods with anthropogenic impacts, and then analyzed these particles using STXM-NEXAFS



at different RH. Substantial changes in particle microstructure were observed upon dehydration,
very likely caused by efflorescence and crystallization of sulfate salts (Pöhlker et al., 2014).
Piens et al. (2016) employed STXM-NEXAFS to examine hygroscopicity of atmospheric
particles collected from the Department of Energy's Atmospheric Radiation Monitoring site in
the Southern Great Plans. As shown in Fig. 11, compared to particles with medium and low
hygroscopicity, particles with high hygroscopicity always contained larger fractions of Na and
Cl (Piens et al., 2016).
**3.3.5 Discussion**
Hygroscopicity measurements using microscopic techniques typically rely on changes in
particle diameter measured microscopically. Therefore, it would be non-trivial for these
techniques to quantify hygroscopic growth factors for non-spherical particles. In addition, these
techniques may not be sensitive enough to investigate water adsorption. Since single particles
deposited on supporting substances are usually examined, these techniques can be employed
to investigate supersaturated samples if proper supporting substances are used. They have also
been widely used to explore hygroscopic properties of ambient aerosol particles which were
collected on proper substances. As discussed in Section 3.4, microscopic techniques can be and
have widely been coupled to spectroscopic tools, and if so chemical information could be
simultaneously provided;
**3.4 Spectroscopic techniques**
Interaction with water vapor would lead to changes in composition and chemical
environment of particles under examination, and these changes can be monitored using
spectroscopic techniques to understand hygroscopic properties of atmospherically relevant
particles.



### 3.4.1 Fourier transform infrared spectroscopy

Fourier transform infrared spectroscopy (FTIR), a vibrational absorption spectroscopy, has been widely employed in laboratory (Goodman et al., 2000; Eliason et al., 2003; Asad et al., 2004b; Hung et al., 2005; Najera et al., 2009; Li et al., 2010; Tan et al., 2016; Tang et al., 2016b) and field work (Maria et al., 2002; Russell et al., 2011; Takahama et al., 2013; Kuzmiakova et al., 2016; Takahama et al., 2016; Takahama et al., 2019) to characterize chemical composition of aerosol particles. It can also be used in aerosol hygroscopicity studies. When water is adsorbed or absorbed by particles, change in IR absorption of particles under investigation due to water uptake can be recorded as a function of RH, and therefore hygroscopic properties of these particles can be characterized. One advantage of FTIR is that it can be coupled with a range of accessories to form different experimental configurations, including transmission FTIR (Cziczo et al., 1997; Braban et al., 2001; Goodman et al., 2001; Zhao et al., 2006; Song and Boily, 2013; Leng et al., 2015; Zawadowicz et al., 2015), attenuated total reflection-FTIR (ATR-FTIR) (Schuttlefield et al., 2007a; Navea et al., 2010; Hatch et al., 2011; Zeng et al., 2014; Zhang et al., 2014a; Yeşilbaş and Boily, 2016; Navea et al., 2017; Gao et al., 2018), diffuse reflectance infrared Fourier transform spectroscopy (DRIFTS) (Gustafsson et al., 2005; Ma et al., 2010a; Joshi et al., 2017; Ibrahim et al., 2018) and micro-FTIR for which FTIR is coupled with a microscope (Liu et al., 2008b; Liu and Laskin, 2009). Particles under investigation are typically deposited on proper substrates, though aerosol particles can also be studied using transmission FTIR (Cziczo et al., 1997; Cziczo and Abbatt, 2000; Zhao et al., 2006; Zawadowicz et al., 2015). FTIR has been used in a large number of studies to investigate hygroscopic properties of atmospherically relevant particles, and herein we only introduce and highlight a few representative examples.

Micro-FTIR was employed to investigate hygroscopic properties of $CH_3SO_3Na$ particles (Liu and Laskin, 2009) and $NH_4NO_3$ (Wu et al., 2007). Fig. 12a shows IR spectra of $CH_3SO_3Na$



particles during humidification, and no significant change in IR spectra was observed when
RH was increased from 0 to 70%; however, when RH was increased to 71%, IR absorption
attributed to the $v(H_2O)$ band (at ~3400 cm$^{-1}$) became very evident and its intensity increased
with further increase in RH, indicating that the deliquescence of $CH_3SO_3Na$ particles occurred
at 71% RH. In addition, at <71% RH two groups of narrow and structured bands, typically
observed for crystalline samples, were observed for $CH_3SO_3Na$ particles. The first one,
centered at ~1197 and 1209 cm$^{-1}$, was attributed to asymmetrical stretching of $v_8(-SO_3^-)$, and
the other one, centered at 1062 cm$^{-1}$, was attributed to symmetrical stretching of $v_3(-SO_3^-)$.
When RH was increased to 71%, both bands were significantly broaden and shifted to lower
wavelengths, further confirming that DRH of $CH_3SO_3Na$ particles was ~71%. IR spectra of
$CH_3SO_3Na$ particles during dehumidification are displayed in Fig. 12b. Complete
disappearance of IR absorption at ~3400 cm$^{-1}$ and significant change in shape and position of
IR peaks of $v_8(-SO_3^-)$ and $v_3(-SO_3^-)$ were observed when RH was decreased from 49 to 48%,
suggesting that the ERH of $CH_3SO_3Na$ was around 48%.



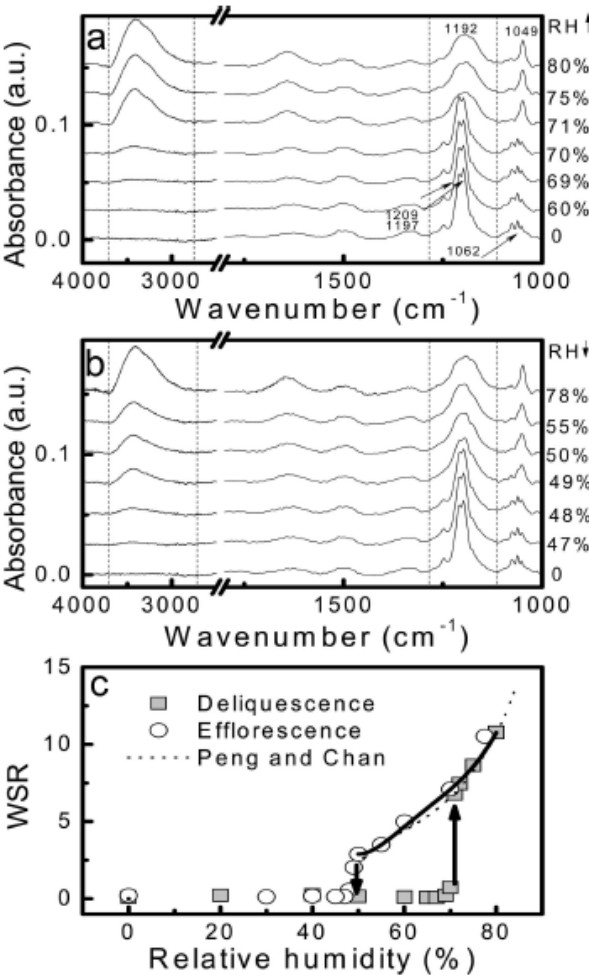


**Figure 12.** (a) FTIR spectra of CH$_3$SO$_3$Na particles during humidification. (b) FTIR spectra of
CH$_3$SO$_3$Na particles during dehumidification. (c) Water-to-solute ratios (WSR) of CH$_3$SO$_3$Na
particles as a function of RH: comparison between WSR measured by Liu and Laskin (2009)
using micro-FTIR to those determined by Peng and Chan (2001b) using electrodynamic
balance. Reprinted with permission by Liu et al. (2009). Copyright 2009 American Chemical
Society.



FTIR spectra can also be used to investigate hygroscopic growth quantitatively if IR
absorbance can be calibrated. In the work by Liu and Laskin (2009), the absorbance ratio of
$v(H_2O)$ (at ~3400 cm$^{-1}$) to $v_8(-SO_3^-)$ (at ~1192 cm$^{-1}$) was calibrated and then used to calculate
water-to-solute ratios (WSR, defined as mole ratios of $H_2O$ to $CH_3SO_3^-$) of aqueous $CH_3SO_3Na$
particles. As shown in Fig. 12c, WSR values determined using FTIR (Liu and Laskin, 2009)
agreed well with those reported in a previous study (Peng and Chan, 2001b) using the
electrodynamic balance (EDB). In another study (Liu et al., 2008b), DRH, ERH and WSR
measured using micro-FTIR were found to agree well with those reported in literature for NaCl,
$NaNO_3$ and $(NH_4)_2SO_4$ particles. ATR-FTIR can be used in a similar way to micro-FTIR to
investigate phase transitions and WSR of atmospherically relevant particles, and has been
applied to a number of compounds, including NaCl (Schuttlefield et al., 2007a; Zeng et al.,
2014), $NaNO_3$ (Tong et al., 2010b; Zhang et al., 2014a), $Na_2SO_4$ (Tong et al., 2010b), $NH_4NO_3$
(Schuttlefield et al., 2007a), $(NH_4)_2SO_4$ (Schuttlefield et al., 2007a), $CH_3SO_3Na$ (Zeng et al.,
2014), sodium formate (Gao et al., 2018), sodium acetate (Gao et al., 2018), and etc.
In addition, ATR-FTIR (Schuttlefield et al., 2007a; Schuttlefield et al., 2007b; Hatch et al.,
2011; Navea et al., 2017), DRIFTS (Ma et al., 2010a; Joshi et al., 2017; Ibrahim et al., 2018)
and transmission FTIR (Goodman et al., 2001) have been employed to investigate water
adsorption by insoluble particles, such as mineral dust. Fig. 13 displays IR spectra of adsorbed
water on $SiO_2$ at different RH, as measured using DRIFTS at 30 °C. As shown in Fig. 13, two
intensive peaks appeared in IR spectra at elevated RH (Ma et al., 2010a), one at 2600-3800
cm$^{-1}$ attributed to the O-H stretching mode and the other one at ~1630-1650 cm$^{-1}$ attributed to
the bending mode of H-O-H. Both peaks can be used to quantify the amount of adsorbed water,
though surface OH groups may also contribute to the IR absorbance at ~3400 cm$^{-1}$ (Goodman
et al., 2001; Tang et al., 2016a). The intensity of the third peak at 2100-2200 cm$^{-1}$, attributed
to the association mode of H-O-H, was much smaller (Ma et al., 2010a). It is possible but non-



trivial to convert IR absorbance to the amount of adsorbed water, and the procedure used can
be found elsewhere (Goodman et al., 2001; Ma et al., 2010a; Joshi et al., 2017; Ibrahim et al.,
2018). It was found that the three-parameter BET equation (Joyner et al., 1945) could well
describe water adsorption as a function of RH on mineral oxides (such as $SiO_2$, $TiO_2$, $Al_2O_3$,
MgO and etc.) (Goodman et al., 2001; Ma et al., 2010a; Joshi et al., 2017), authentic mineral
dust from different sources (Joshi et al., 2017; Ibrahim et al., 2018) and Icelandic volcanic ash
(Joshi et al., 2017). Another study (Hatch et al., 2011) suggested that compared to the two-
parameter BET equation, the Freundlich adsorption isotherm could better approximate the
amount of water adsorbed by kaolinite, illite, and montmorillonite at different RH.

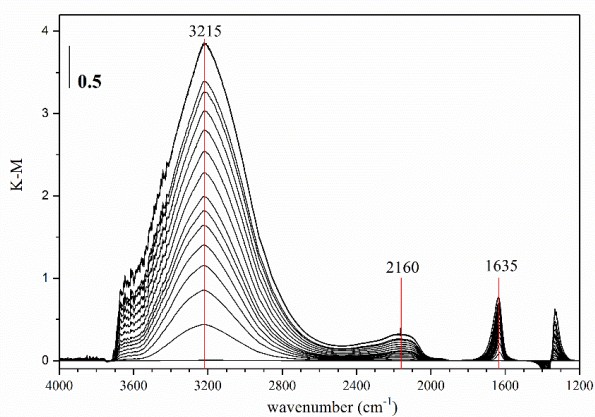


**Figure 13.** IR spectra of adsorbed water on $SiO_2$ at 30 °C, as measured using DRIFTS at
different RH. Reprint (with modification) with permission by Ma et al. (2010a). Copyright
2011 Elsevier.

**3.4.2 Raman spectroscopy**

Raman spectroscopy is complementary to infrared spectroscopy. Bands which are weak in

infrared spectroscopy can be strong in Raman spectroscopy, and vice versa. Compared to
infrared spectroscopy, Raman spectroscopy is much less sensitive to $H_2O$, despite that
symmetric stretching vibration of $H_2O$ is Raman active, and this characteristic limits





application on Raman spectroscopy in exploring particles with low hygroscopicity. Meanwhile,
Raman spectroscopy is very sensitive to crystalline structures, making it very useful to
investigate particle phase transition. For example, Raman spectroscopy was employed to probe
phase transformation of levitated $(NH_4)_2SO_4$, $Na_2SO_4$, $LiClO_4$, $Sr(NO_3)_2$, $KHSO_4$, $RbHSO_4$
and $NH_4HSO_4$ microparticles (Tang et al., 1995), and the occurrence of metastable solid states
was observed under ambient conditions for $Na_2SO_4$, $LiClO_4$, $Sr(NO_3)_2$ and bisulfates. Raman
spectroscopy was also used to investigate hygroscopic properties of supersaturated droplets
(Zhang and Chan, 2000; Zhang and Chan, 2002b), such as $(NH_4)_2SO_4$ and $MgSO_4$.

For regular spherical droplets, their Raman spectra may overlap with strong morphology-

dependent resonances (Zhang and Chan, 2002b). Nevertheless, if individual droplets were
deposited on proper substrates, Raman spectra with high quality (i.e. high signal-to-noise ratios)
could be obtained using confocal micro-Raman spectroscopy (Wang et al., 2005; Li et al.,
2006). For example, micro-Raman spectrometry was successfully used to investigate
hygroscopic properties of $(NH_4)_2SO_4$, $Ca(NO_3)_2$ and $NO_2$-aged $Ca(NO_3)_2$ particles deposited
on fluorinated ethylene propylene slides (Liu et al., 2008c; Zhao, 2010). Herein we use
$(NH_4)_2SO_4$ as an example to illustrate how Raman spectroscopy can be used to determine
hygroscopic properties of atmospherically relevant particles. Fig. 14 shows Raman spectra and
microscopic images of an $(NH_4)_2SO_4$ particle at different RH during humidification and
dehumidification processes (Zhao, 2010). When RH was increased to 80% during
humidification, the Raman peak centered at ~3450 cm$^{-1}$, attributed to the stretching vibration
of $H_2O$, started to become evident; whereas during dehumidification this peak disappeared
when RH was decreased to 37%. This suggested that deliquescence and efflorescence of
$(NH_4)_2SO_4$ took place at 80 and 37% RH, respectively.



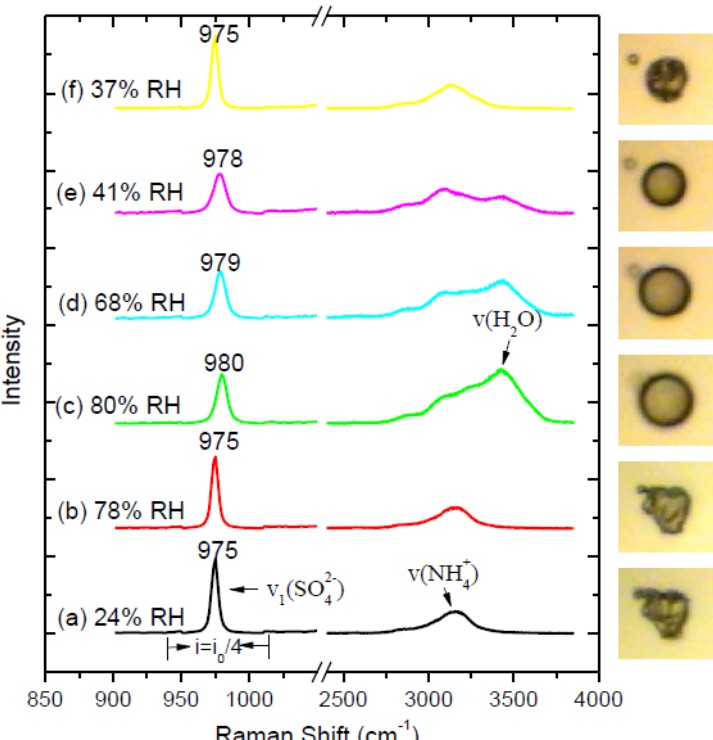


**Figure 14.** Raman spectra and microscopic images of an $(NH_4)_2SO_4$ particle during

humidification (a-c) and dehumidification (c-f). Reprint with permission by Liu (2008).

Copyright 2008 Peking University.


As discussed in previous work (Ling and Chan, 2007; Liu et al., 2008c; Zhao, 2010), the

occurrence of deliquescence and efflorescence of $(NH_4)_2SO_4$ could also be identified from the

change in position and full width at half maxima (FWHM) of the Raman peak at 970-980 cm$^{-1}$

(due to symmetrical stretching of sulfate, $v_1$-$SO_4^{2-}$). As shown in Fig. 14, during humidification

$v_1$-$SO_4^{2-}$ was shifted from 975 to 980 cm$^{-1}$ when RH was increased to 80%, and meanwhile its

FWHM increased from 6 to 9 cm$^{-1}$, implying the occurrence of deliquescence. For comparison,

during dehumidification when RH was decreased to 37%, $v_1$-$SO_4^{2-}$ was shifted from 978-980

to 975 cm$^{-1}$ and the corresponding FWHM decreased from ~10 to 6 cm$^{-1}$, suggesting that





efflorescence took place at ~37% RH. Phase transitions could be further inferred from
microscopic images (Liu et al., 2008c; Zhao, 2010). Fig. 14 shows that the particle under
investigation became spherical when it was deliquesced (at 80% RH), and became irregular
when efflorescence occurred (at ~37% RH).

The peak intensity ratio of stretching vibration of $H_2O$ to symmetrical stretching of sulfate

is proportional to the molar ratio of $H_2O$ to sulfate in the solution, and could be used to quantify
the water-to-solute ratios (WSR) in aqueous $(NH_4)_2SO_4$ droplets if properly calibrated (Liu et
al., 2008c). WSR values determined using Raman spectroscopy (Liu et al., 2008c) were found
to agree well with those reported in literature as a function of RH for $(NH_4)_2SO_4$ and $Ca(NO_3)_2$
during humidification and dehumidification processes (Stokes and Robinson, 1948; Tang and
Munkelwitz, 1994; Clegg et al., 1998; Kelly and Wexler, 2005). In addition, Liu et al. (2008c)
employed micro-Raman spectroscopy to study heterogeneous reaction of $CaCO_3$ with $NO_2$,
and revealed that solid $CaCO_3$ particles were converted to aqueous droplets after heterogeneous
reaction with $NO_2$, due to the formation of $Ca(NO_3)_2$.

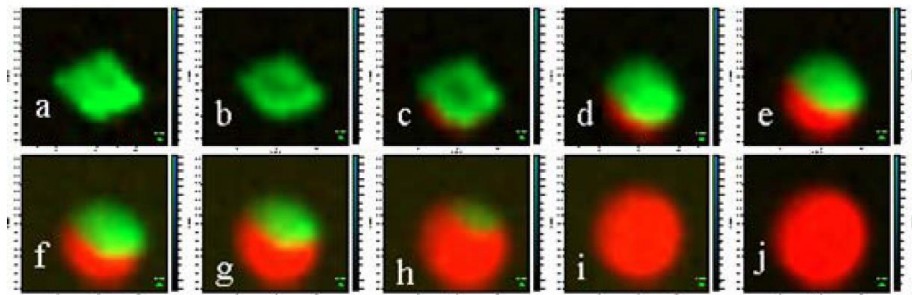


**Figure 15.** Spatial distribution of nitrate (red) and carbonate (green) for a $CaCO_3$ particle
during heterogeneous reaction with 50 ppmv $NO_2$ at 40% RH. The reaction time was 0 (a), 10
(b), 23 (c), 38 (d), 58 (e), 64.5 (f), 77.5 (g), 105 (h), 119 (i) and 126 min (j). Reprint with
permission by Zhao et al. (2010). Copyright 2010 Peking University.




Raman microscopy was further used to map spatial distribution of chemical composition
of individual $CaCO_3$ particles during heterogeneous reaction with 50 ppmv $NO_2$ at 40% RH
(Zhao, 2010), and the results are displayed in Fig. 15-16. As the reaction proceeded, the spatial
coverage decreased with time for carbonate and increased for nitrate (Fig. 15), suggesting that
$CaCO_3$ was converted to $Ca(NO_3)_2$ upon exposure to $NO_2$. Meanwhile, the spatial coverage of
particle water also increased with time (Fig. 16), and its spatial distribution overlapped well
with that of nitrate, suggesting that the increase in particle water content was caused by the
formation of hygroscopic $Ca(NO_3)_2$. In addition, microscopic images revealed that the particle
size increased with reaction time, suggesting that hygroscopicity of the particle under
investigation increased as heterogeneous reaction with $NO_2$ proceeded.

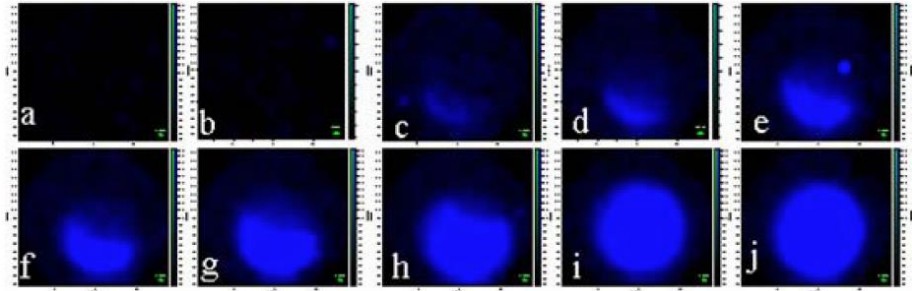


**Figure 16.** Spatial distribution of particle water content for a $CaCO_3$ particle during
heterogeneous reaction with 50 ppmv $NO_2$ at 40% RH. The reaction time was 0 (a), 10 (b), 23
(c), 38 (d), 58 (e), 64.5 (f), 77.5 (g), 105 (h), 119 (i) and 126 min (j). Reprint with permission
by Zhao et al. (2010). Copyright 2010 Peking University.

Raman spectroscopy has been employed in a number of studies to investigate hygroscopic
properties of organic aerosols and mixed particles (Ling and Chan, 2007; Ling and Chan, 2008;
Yeung et al., 2009; Yeung and Chan, 2010; Yeung et al., 2010; Ma and He, 2012; Ma et al.,
2013a; Ma et al., 2013b). During humidification-dehumidification processes, oxalic acid was
converted to oxalate when mixed with NaCl (Ma et al., 2013b) or $Ca(NO_3)_2$ (Ma and He, 2012),



and such conversion would lead to significant change in hygroscopic properties of mixed
particles. When a hygroscopic sulfate, such as $(NH_4)_2SO_4$ or $Na_2SO_4$), was mixed with a
hygroscopic calcium salt, such as $Ca(NO_3)_2$ or $CaCl_2$, gypsum, the hygroscopicity of which
was very limited, would be formed by humidification. Raman spectroscopy was also used to
explore hygroscopic properties of $NH_4NO_3/(NH_4)_2SO_4$ mixed particles (Ling and Chan, 2007),
and    the    formation    of    double-salts,    including    $3(NH_4NO_3)\cdot(NH_4)_2SO_4$    and
$2(NH_4NO_3)\cdot(NH_4)_2SO_4$, was observed for the first time during crystallization. The effect of
malonic, glutaric and succinic acids on the hygroscopic properties of $(NH_4)_2SO_4$ particles were
explored using Raman spectroscopy (Ling and Chan, 2008). Partial crystallization of
$(NH_4)_2SO_4$/malonic acid droplets took place at 16% RH, while $(NH_4)_2SO_4$/glutaric acid and
$(NH_4)_2SO_4$/succinic acid particles became completely effloresced at ~30% RH. In addition,
partial deliquescence with solid inclusions was observed at 10-79% RH for $(NH_4)_2SO_4$/malonic
acid, 70-80% for $(NH_4)_2SO_4$/glutaric acid, and 80-90% RH for $(NH_4)_2SO_4$/succinic acid
particles.
**3.4.3 Fluorescence spectroscopy**

Water molecules in aqueous solutions can exist in two states, i.e. solvated water which

interacts directly with ions, and free water which interacts with other water molecules. Chan
and co-workers (Choi et al., 2004; Choi and Chan, 2005) developed a method to explore the
state of water molecules in single droplets levitated in an EDB. Pyranine, a water soluble dye,
was added into the droplets. When excited by radiation at ~345 nm, Pyranine would emit
fluorescence, and the spectra peaked at ~440 nm (attributed to the presence of solvated water)
and ~510 nm (attributed to the presence of free water). The amounts of solvated and free water
can be derived by combining mass hygroscopic growth factors (determined using the EDB)
and the ratio of fluorescence intensity at 440 nm to that at 510 nm (Choi et al., 2004). It was
found that for NaCl, $Na_2SO_4$ and $(NH_4)_2SO_4$, efflorescence of supersaturated droplets occurred



when the amount of solvated water was equal to that of free water (Choi et al., 2004; Choi and
Chan, 2005). Imaging analysis further revealed that solvated and free water were
homogeneously distributed in the droplets for some types of droplets, e.g., $MgSO_4$, but
heterogeneously distributed for other types of droplets, such as NaCl and $Na_2SO_4$ (Choi and
Chan, 2005).
In another study (Montgomery et al., 2015), fluorescence microscopy was used to monitor
structural change of particle aggregates with RH. In this work NaCl particle aggregates were
collected on wire meshes and then coated with Rhodamine which would generate fluorescence.
Particle aggregates collapsed and became more compact when RH was increased from 0 to 52%
(Montgomery et al., 2015), lower than the DRH of NaCl (~75%). Hosny et al. (2013) developed
fluorescence lifetime imaging microscopy (FLIM) to determine viscosity of individual
particles via measuring viscosity dependent fluorescence lifetime of fluorescent molecular
rotors. The viscosity of a particles is of interest because it is closely related to the phase state
of the particle and largely determines diffusion in the particle (Koop et al., 2011; Reid et al.,
2018). FLIM was used to investigate the viscosity of ozonated oleic acid particles and
secondary organic particles formed by myrcene ozonolysis, and their viscosity was observed
to increase largely with decreasing RH and increasing extent in oxidative aging (Hosny et al.,

2016).

### 3.4.4 Other surface characterization techniques

In addition to spectroscopic and microscopic methods discussed in Sections 3.3 and 3.4,
there are a number of other surface characterization techniques which can be used to explore
water adsorption on surfaces, e.g., sum frequency generation spectroscopy (Ma et al., 2004;
Liu et al., 2005; Jubb et al., 2012; Ault et al., 2013), atmospheric pressure X-ray photoelectron
spectroscopy (Ketteler et al., 2007; Salmeron and Schlogl, 2008; Yamamoto et al., 2010a),
scanning tunneling microscopy (Wendt et al., 2006; He et al., 2009), and etc. These techniques,



which are able to provide fundamental and mechanistic insights into water-surface interactions,
have mainly been applied to surfaces of single crystals, and their usefulness for particles with
direct atmospheric relevance is yet to be demonstrated. As a result, these techniques are not
further discussed here, and readers are referred to aforementioned literature and references
therein for more details.
**3.4.5 Discussion**
Infrared and Raman spectroscopy can be used to quantify particle water content for
unsaturated and supersaturated samples, with no restriction imposed by particle shape or
morphology. Infrared spectroscopy is very sensitive to adsorbed water and has been widely
used to investigate water adsorption (Tang et al., 2016a), as discussed in Section 3.3.1. In
contrast, Raman spectroscopy is not sensitive enough to detect adsorbed water; nevertheless,
recent work (Gen and Chan, 2017) showed that electrospray surface enhanced Raman
spectroscopy was able to detect surface adsorbed water. One important advantage for infrared
and Raman spectroscopy is that simultaneous measurement of chemical composition can be
provided; therefore, they have been coupled to other techniques (such as optical microscope,
electrodynamic balance, and etc.) to further understand hygroscopic properties of
atmospherically relevant particles, as discussed in Sections 3.3, 3.4, 4.1 and 4.2. Infrared and
Raman spectroscopy have been widely employed to characterize ambient aerosol particles
collected on proper substrates, and therefore they can be used to explore hygroscopic properties
of ambient particles in an offline manner.
**3.5 Measurement of electrical properties**
Deliquescence of ionic solids would lead to significant increase in electrical conductivity
and vice versa efflorescence of electrolyte solutions to ionic solids would cause large decrease
in electrical conductivity. Therefore, relative changes in electrical conductivity/impedance can
be used to identify the occurrence of deliquescence and efflorescence (Yang et al., 2006; He et
al., 2008; Schindelholz et al., 2014a; Schindelholz et al., 2014c). For example, in one study
(Schindelholz et al., 2014c) micrometer-sized particles were deposited on an interdigitated
microelectrode sensor housed in an environmental chamber, and the electrical impedance was
detected online while RH in the chamber was varied. The measured DRH and ERH using this
method were found to agree well with literature values for several compounds, e.g., NaCl,
NaBr and KCl (Schindelholz et al., 2014c). In another study (He et al., 2008), the electrical
conductivity and capacitance of a single droplet were measured as different RH to investigate
hygroscopic properties of $NaClO_4$ particles. Overall, this method has not been widely applied
to study atmospherically relevant particles and thus is not further discussed herein.
**4 Levitated single particles**
Single particle levitation techniques can be broadly classified into three groups (Krieger et
al., 2012), including electrodynamic balance, optical levitation and acoustic levitation. These
techniques have been widely used to investigate chemical and physical transformation of
atmospherically relevant particles (Lee et al., 2008; Krieger et al., 2012). Herein we introduce
basic principles of each techniques and illustrate how they can help understand aerosol
hygroscopicity via discussing representative studies.
**4.1 Electrodynamic balance**
The electrodynamic balance (EDB) technique has been widely used in the last several
decades, and diameters of particles which can be levitated by EDB are typically in the range of
1-100 μm (Davis, 1997; Davis, 2011). The principle, configuration and operation of EDB have
been extensively documented elsewhere (Reid and Sayer, 2003; Lee et al., 2008; Davis, 2011;
Krieger et al., 2012), and hence are not described in detail here. In brief, a particle can be
levitated and trapped at the null point of the EBD chamber when the AC and DC electric fields
surrounding the particle are properly adjusted. The schematic diagram of a low-temperature
EDB (Tong et al., 2015) is shown in Fig. 17. The main body of the EDB was an octagonal





aluminum chamber with an optical window on each side. Two cold nitrogen flows, which were
first passed through copper tubes immersed in a liquid nitrogen Dewar, were fed into the
chamber to cool the EDB. Temperature at the null point where a particle was trapped was
further regulated using a PTC heater, and temperature and RH inside the chamber were
monitored online. A continuous-wave laser at 532 nm was used to illuminate the trapped
particle, and the scattered light was measured at an angle of 21° to determine the particle size.

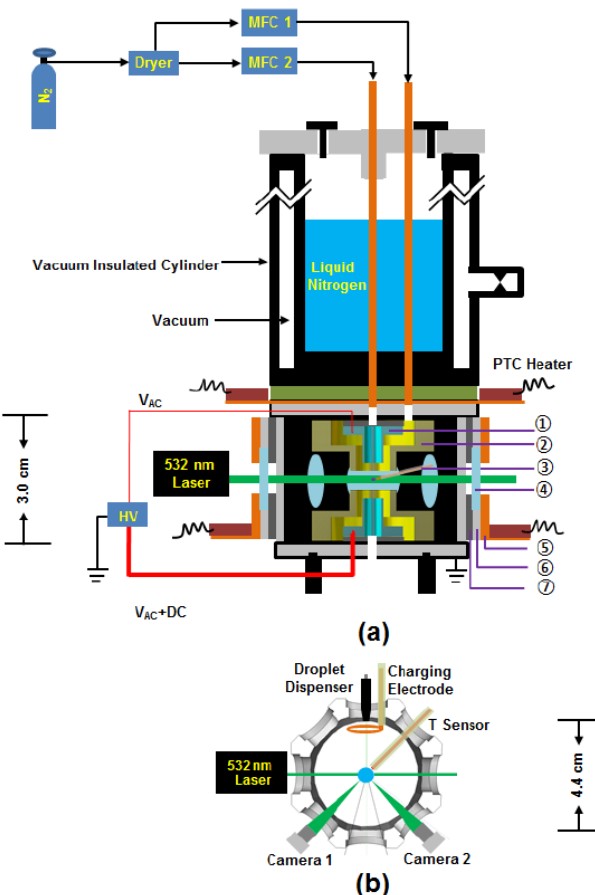


**Figure 17.** Schematic diagram of a cold electrodynamic balance. (a) Side view of this set-up:
1) inner electrode; 2) outer electrode; 3) temperature and RH sensors; 4) glass optical window;
5) heating jacket; 6) optical window holder; 7) rubber insulator. (b) Top view of this set-up:
droplets were generated using a droplet dispenser and charged using a charging electrode, and



one of them may be trapped at the null point. A 532 nm laser was used to illuminate the trapped
particle, and two cameras were used to observe the particle and record the scattered light.
Reprint with permission by Tong et al. (2015). Copyright 2015 Copernicus Publications.

In the absence of other forces, the gravitational force of the particle trapped in the EDB is

equal to the balancing electrostatic force, given by Eq. (2) (Pope et al., 2010a; Davis, 2011):
$$mg = nqC\frac{V_{DC}}{z} \quad (2)$$

where $m$ is the particles mass, $g$ is the gravitational constant, $n$ is the number of elementary
charges present on the particles, $q$ is the elementary charge, $z$ is the distance between the two
electrodes, $C$ is a constant dependent on the geometrical configuration of the EDB, and $V_{DC}$ is
the DC voltage required to levitate the particle. Eq. (2) suggests that as long as the charge
present on the trapped particle remains constant, the mass of the particle is proportional to the
DC voltage required to balance its gravitational force. Therefore, the relative mass change of
the particle due to any physical or chemical processing can be quantified by measurement of
the DC voltage. Haddrell et al. (Haddrell et al., 2012) discussed conditions when the
assumption of constant charge may fail and proposed experimental strategies to minimize its
occurrence.

In hygroscopicity studies, the relative mass change of the trapped particle (typically

relative to that under dry condition) during humidification and dehumidification can be
determined to obtain mass hygroscopic growth factors (Peng et al., 2001; Pope et al., 2010a;
Haddrell et al., 2014; Steimer et al., 2015). For example, EDB has been used to measure DRH,
ERH and mass hygroscopic growth factors for a number of inorganic (Tang and Munkelwitz,
1994; Tang and Fung, 1997; Tang et al., 1997; Zhang and Chan, 2002a; Zhang and Chan, 2003;
Hargreaves et al., 2010b), organic (Peng and Chan, 2001a; Peng et al., 2001; Choi and Chan,
2002a; Pope et al., 2010a; Steimer et al., 2015) and mixed inorganic/organic particles (Choi



and Chan, 2002b; Zardini et al., 2008; Pope et al., 2010a) of atmospheric relevance. In addition,
water uptake by different types of pollen was measured as a function of RH using  an EDB
(Pope, 2010; Griffiths et al., 2012). As displayed in Fig. 17, pollen grains were found to be
moderately hygroscopic, and the mass of water taken up at 90% RH was around 30% of the
dry mass (Pope, 2010). It was further found that hygroscopic growth of pollen species could
be described by the $\kappa$-Kohler theory, with $\kappa$ values falling in the range of 0.05-0.1 (Pope, 2010).
In another two studies (Haddrell et al., 2013; Haddrell et al., 2014), EDB was utilized to explore
hygroscopic growth of several pharmaceutically relevant formulations, and the results can help
better understand where medical aerosol particles would deposit in our inhalation system.

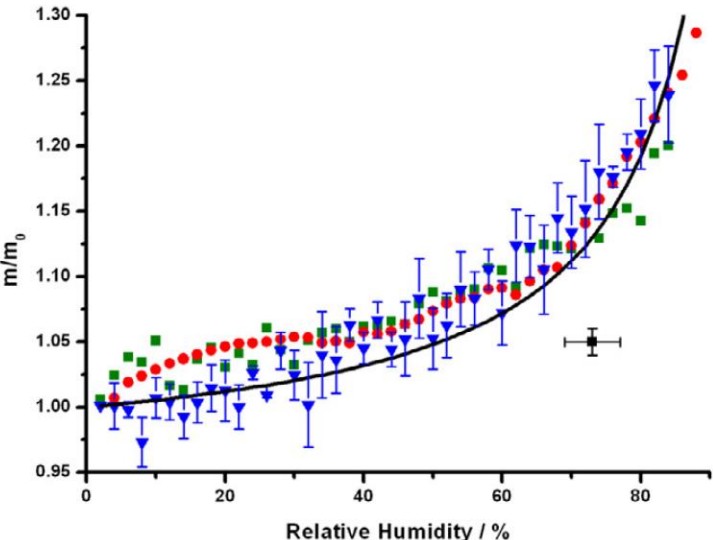


**Figure 18.** Mass hygroscopic growth factors (defined as the ratio of the particle mass at a given
RH to the dry particle mass) of Salix caprea (red circle), Betula occidentalis (blue triangle),
and Narcissus sp. (green square). For clarity only the error bars (±1 σ) are shown for Betula
occidentalis, and the mass hygroscopic grow factors have similar uncertainties for the other
two pollen species. The black square represents water uptake reported by Diehl et al. (2001),
and the black curve represents the fitted mass hygroscopic growth curve using the $\kappa$-Kohler
theory. Reprint with permission by Pope (2010). Copyright 2010 IOP Publishing Ltd.




Light scattering techniques can used to measure optical properties of single particles
levitated in an EDB. For example, Tang and co-workers (Tang and Munkelwitz, 1994; Tang,
1997; Tang and Fung, 1997; Tang et al., 1997) measured the intensity of elastically scattered
light from a levitated particle which was illuminated by a He-Ne laser beam, and managed to
retrieve its diameter and refractive index as a function of RH using the Mie theory. Since the
relative mass change was also determined at the same time, change in particle density with RH
could also be determined (Tang and Munkelwitz, 1994; Tang et al., 1997). In addition,
spectroscopic techniques have been frequently coupled to EDB in order that chemical
information could be simultaneously provided. For example, Chan and colleagues (Zhang and
Chan, 2002a; Zhang and Chan, 2003; Lee et al., 2008) directed a laser beam with a wavelength
of 514.5 nm to the trapped particle in the EDB and measured the resulting Raman signals with
a CCD detector. This configuration enabled change in particle composition and hygroscopicity
due to heterogeneous reactions to be monitored online in a simultaneous manner (Lee and Chan,
2007; Lee et al., 2008). Experimental work in which EDB was coupled to fluorescence
spectroscopy has also been reported (Choi et al., 2004; Choi and Chan, 2005).
In addition to hygroscopicity research, EDB have also been used in a number of studies
(Reid and Sayer, 2003; Lee et al., 2008; Pope et al., 2010b; Davis, 2011; Krieger et al., 2012;
Bilde et al., 2015b) to investigate other physicochemical properties (including vapor pressure,
mass accommodation coefficients, evaporation coefficients, gas phase diffusion coefficients,
and etc.) and chemical reactions of atmospheric particles.
**4.2 Optical levitation**
Trapping and manipulation of atoms, molecules, nanostructures and particles have been
widely used in a number of scientific fields (Ashkin, 2000; McGloin, 2006; Mitchem and Reid,
2008; Krieger et al., 2012; Lehmuskero et al., 2015; Spesyvtseva and Dholakia, 2016; Gong et



al., 2018). The effects of radiation pressure on microscopic particles were first demonstrated
in 1970 (Ashkin, 1970). After that, levitation of solid particles and liquid droplets in air using
a vertically propagating weakly focused laser beam was achieved (Ashkin and Dziedzic, 1971;
Ashkin and Dziedzic, 1975). Applications of optical levitation to particles of atmospheric
relevance have been previously reviewed (Mitchem and Reid, 2008; Wills et al., 2009; Krieger
et al., 2012), and very recently general applications related to trapping single particles in air
have also been summarized (Gong et al., 2018).
Interaction of an incident laser beam with a particle consists of two forces: (i) a scattering
force that results from the transfer of momentum to the dielectric particle from backscattered
photons, and (ii) a gradient force that depends on the gradient of the electromagnetic field
associated with the optical beam. The first type of force exerts a push on the particle, while the
second type exerts a pull (Krieger et al., 2012). Utilization of either of these two forces as the
primary force to trap particles leads to two types of optical levitation techniques, i.e. optical
levitation trap and optical tweezers. In an optical levitation trap, the laser beam is mildly
focused and the particle adopts a stable position within the divergent beam above the focus,
where the downward gravitational force is exactly balanced by the upward scattering force
(Wills et al., 2009). Droplets of 20-100 µm in diameter can be trapped with active
compensating adjustment of light intensity with respect to changes in droplet size (Krieger et
al., 2012); nevertheless, optical levitation traps are intrinsically delicate and unstable (Wills et
al., 2009). Optical tweezers effectively create a strong intensity gradient in three dimensions,
by amplifying the gradient force using a microscope objective lens to tightly focus the trapping
laser beam. The gradient force leads to strong transverse and axial restoring forces that are
many orders of magnitude larger than the gravitational force of the particle (Wills et al., 2009),
restoring the particle to the region with the highest light intensity (Krieger et al., 2012).
Therefore, particles can be captured and held tightly against the scattering and gravitational



forces, allowing true 3-dimensional confinement of particles with diameters of 1-10 µm
(Krieger et al., 2012).
Different laser beams have been used as incident light sources. In optical levitation traps,
mildly focused Gaussian beams (Ashkin and Dziedzic, 1975), counter-propagating Gaussian
beams (Ashkin, 2000) and a Gaussian beam plus a Bessel beam (Davis et al., 2015a) can be
used to trap single particles. In optical tweezers, particles can be trapped with a single laser
beam (Magome et al., 2003; Mitchem et al., 2006a) or in a dual-trap configuration with two (or
split) laser beams (Fallman and Axner, 1997; Buajarern et al., 2006; Butler et al., 2008), and
counter-propagating Bessel beams have also been used (Lu et al., 2014). Fig. 19 shows a typical
experimental setup for a dual-trap configuration of optical tweezers in which droplets were
generated using a nebulizer and then introduced into the trapping cell (Butler et al., 2008). A
laser beam at 532 nm was used as the trapping light and focused by an oil immersion objective
to create a working distance of ~130 µm. A beam splitter was then used to create two parallel
trapping beams that could be translated independently over distances of >50 µm, allowing
individual manipulation or probing of two separate particles in close range.





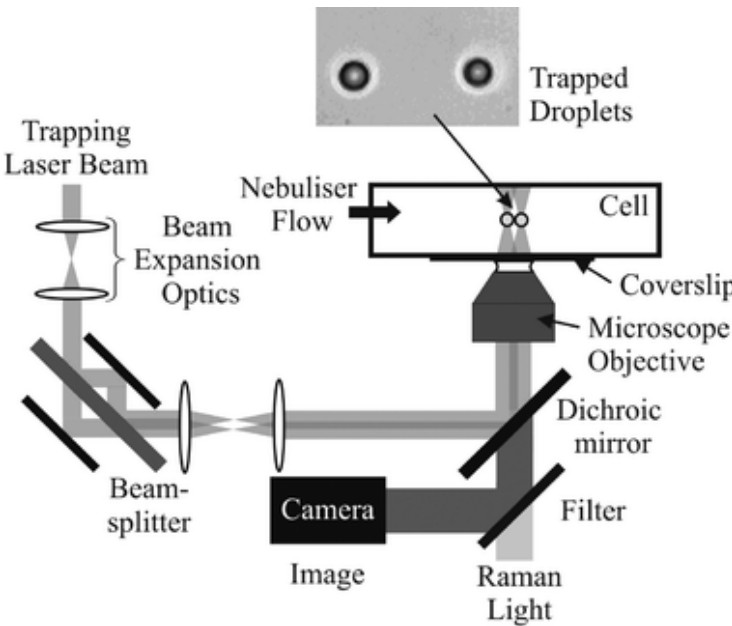


**Figure 19.** Schematic diagram of the dual trap configuration of the optical tweezers. Reprint
with permission by Butler et al. (2008). Copyright 2008 Royal Society of Chemistry.

When a single particle is optically trapped, it can be characterized by a number of

techniques. Direct imaging is the most straightforward one, and bright field imaging can be
used to determine particle size with an accuracy of ±0.2 µm (Burnham and McGloin, 2009).
However, this method suffers from low accuracy in size measurement due to the dependence
of the axial position on laser power (Knox et al., 2007). Spectroscopy, especially Raman
spectroscopy, is more accurate in particle size measurement (Wills et al., 2009) and can also
offer compositional information (Reid et al., 2007). Known as cavity-enhanced Raman
spectroscopy, spectra recorded from optically trapped particles comprise of spontaneous and
stimulated Raman scattering (Mitchem et al., 2006a; Wills et al., 2009). Spontaneous Raman
scattering can be used to investigate changes in OH stretching vibrations (2900-3700 cm$^{-1}$) of
particulate water during hygroscopic growth as well as hydrogen bonding environments within




the particle. On the other hand, stimulated Raman scattering can be strongly amplified (by a
factor of >10) (Mitchem et al., 2006a), but it occurs only at distinct wavelengths that are
commensurate with whispering gallery modes (WGMs). This stimulated Raman scattering
under WGMs, as shown in Fig. 20, is also commonly referred to as morphology-dependent
resonances or cavity resonances (Mitchem et al., 2006a). Using the stimulated Raman spectra,
one can achieve a sizing accuracy of ±2 nm that is only limited by spectral dispersion of the
spectrograph (Mitchem et al., 2006a; Mitchem and Reid, 2008). Other techniques have also
been coupled with optical levitation, including elastic (Mie) scattering (Ward et al., 2008), light
absorption (Knox and Reid, 2008), and so on.

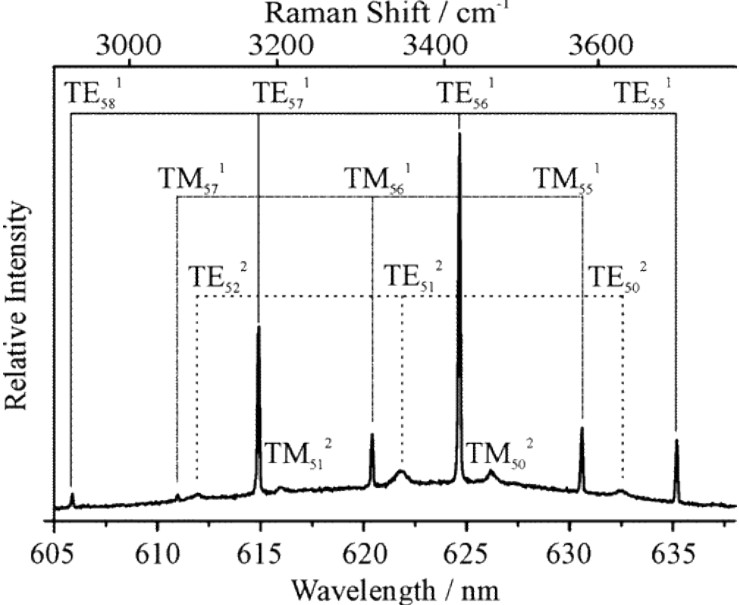


**Figure 20.** An example of Raman scattering from a trapped water droplet, illuminated at 514.5
nm. Stimulated Raman scattering is observed at wavelengths commensurate with whispering
gallery modes. The resonant modes can be assigned by comparison with Mie scattering
calculations, and the droplet radius can then be derived. Reprint with permission by Mitchem
et al. (2006a). Copyright 2006 American Chemical Society.



There are a number of studies in which optical levitation techniques were employed to
investigate hygroscopic properties of atmospheric particles. Based on an early design (Hopkins
et al., 2004), Mitchem et al. (2006a) investigated hygroscopic growth of a NaCl particle trapped
by optical tweezers for RH >80% by characterizing spontaneous and stimulated Raman
scattering. Changes in the OH stretching band of the particle were observed as RH increased,
and size measurement was achieved with an accuracy of a few nanometre and a time resolution
of 1 s. The measured equilibrium sizes agreed well with these predicted using the Köhler theory,
and the largest uncertainties came from the error in RH measurement with a capacitive sensor
(±2% for RH below 90%) (Mitchem et al., 2006a). The change in the OH stretching band was
also used to probe the formation and destruction of hydrogen bonding in a trapped NaCl particle
at different RH (Treuel et al., 2010).
A dual-trap configuration of optical tweezers, in which two particles could be levitated
simultaneously (as shown in Fig. 19), was employed to investigate hygroscopic properties of
individual particles (Butler et al., 2008). In this setup, the first particle with well-known
hygroscopicity (in this case, NaCl) served as an accurate RH probe (±0.09% even for
RH >90%), while the second particle (NaCl/glutaric acid, for example) was interrogated for its
hygroscopic properties as an "unknown" particle. Excellent agreement between experimental
measurement and prediction using the Köhler theory was achieved (Butler et al., 2008).
Hygroscopic properties of inorganic/organic mixed particles, including NaCl/glutartic acid and
$(NH_4)_2SO_4$/glutartic acid mixtures with different mass ratios, were further studied using this
comparative approach (Hanford et al., 2008). Measured equilibrium sizes of those
inorganic/organic mixed particles were found to agree well with theoretical predictions,
demonstrating the robustness of this approach for hygroscopicity study at the high RH (>97%).
Using the dual-trap configuration, hygroscopic properties of NaCl and $(NH_4)_2SO_4$ were
measured at low RH (down to 80%) (Walker et al., 2010). The usage of NaCl as a reference



particle could reduce the errors associated with the measured equilibrium wet size of
$(NH_4)_2SO_4$ to <0.2%; for comparison, the errors could be as large as ±5% when a capacitance
RH probe was used. The difference between the measured and modelled growth factors was
found to be in the range of 0.1-0.3% for $(NH_4)_2SO_4$ in the medium RH region (84 − 96% RH)
(Walker et al., 2010). In a following study (Hargreaves et al., 2010a), the dual-trap
configuration was utilized to investigate hygroscopic properties of NaCl at 45-75% RH, and
growth factors of NaCl measured by this (Hargreaves et al., 2010a) and previous studies (Butler
et al., 2008; Hanford et al., 2008) were found to be in excellent agreement with those predicted
(Clegg and Wexler, 2011) for RH in the range of 45-99%.

Optical levitation can also be used to explore phase transitions and surface hydration. For

example, liquid to solid phase transitions were observed for the $(NH_4)_2SO_4$/glycerol/$H_2O$
system via morphology-dependent resonances and Raman spectroscopy (Trunk et al., 1997),
and Raman spectroscopy revealed the presence of adsorbed water on the surface of optically
levitated mineral oxide particles at different RH (Rkiouak et al., 2014). In addition, optical
tweezers were utilized to investigate efflorescence and deliquescence of a number of inorganic
salts (Davis et al., 2015a). Compared to deliquescence, efflorescence usually occurs for a lower
RH (Martin, 2000). Immersion of solid particles (e.g., mineral dust) in aqueous droplets would
cause efflorescence to take place at higher RH, as observed in previous work (Han et al., 2002;
Pant et al., 2006). Recently optical levitation was employed to explore efflorescence of
supersaturated aqueous droplets induced by collision with solid particles (Davis et al., 2015a;
Davis et al., 2015b). It was found that upon collision with several different types of solid
particles, including NaCl, KCl, $(NH_4)_2SO_4$, $Na_2SO_4$, and etc., aqueous $NH_4NO_3$, $(NH_4)_2SO_4$
and NaCl droplets would effloresce at RH significantly higher than those for homogeneous
efflorescence (Davis et al., 2015b).



Kinetics of water uptake by aerosol particles can also be studied using optical levitation
techniques. For example, hygroscopic properties of NaCl particles coated with oleic acid was
examined using optical tweezers (Dennis-Smither et al., 2012). It was observed that
efflorescence and deliquescence behavior of the NaCl particle and the timescales to reach re-
equilibrium were not affected by the presence of oleic acid; furthermore, heterogeneous
oxidation by $O_3$ was found to increase the hygroscopicity of oleic acid in the NaCl-oleic acid
mixed particle (Dennis-Smither et al., 2012). In another study (Tong et al., 2011), optical
tweezers were employed to explore the timescales for mass transfer of water in glassy aerosol
particles. It was found that the half-time for re-equilibration after RH change could increase
from tens and hundreds of seconds (RH above glass transition) to >1000 seconds (RH below
glass transition) for sucrose-water, raffinose-water and sucrose-NaCl-water systems.
Particle viscosity determines diffusion coefficients of water molecules in the particles,
affecting water uptake kinetics (Reid et al., 2018). A novel microrheological method, which
employed holographic aerosol optical tweezers, has been developed to measure particle
viscosity in the range of $10^{-3}$ to $10^9$ Pa S (Power et al., 2013). In brief, coalescence between
two airborne particles, with volumes smaller than 500 femtolitres, was initiated using the
optical tweezers, and the time required by the coalesced particle to relax to a sphere was
measured to infer particle viscosity. More details of this method can be found elsewhere (Power
et al., 2013; Song et al., 2016).
In addition, optical levitation techniques have also been employed to investigate a myriad
of heterogeneous processes, including evaporation of volatile/semi-volatile species, mixing of
inorganic/organic particles and heterogeneous reactions (Mitchem et al., 2006b; Buajarern et
al., 2007; Tang et al., 2014; Jones et al., 2015; Gorkowski et al., 2016; Cai and Zhang, 2017).
Optical tweezers have recently become commercially available, and commercial instruments



have been used to investigate physicochemical properties and processes of atmospherically
relevant particles (Davies and Wilson, 2016; Haddrell et al., 2017).

**4.3 Acoustic levitation**

Inside a typical acoustic levitator, high frequency sound wave, generated using a
piezoelectric oscillator (also called radiator), is reflected by a concave reflector. Standing
waves can be generated in the space between the radiator and the reflector if the radiator and
the reflector are properly positioned. Droplets with diameters ranging from tens of micrometers
to a few millimeters can then be trapped in the vertical positon near one of these existing wave
nodes. Detailed description of this technique can be found elsewhere (Kavouras and Krammer,
2003b; Ettner et al., 2004; Mason et al., 2008). The size of the levitated particle can be
characterized using a camera, and spectroscopic techniques, such as FTIR and Raman
spectroscopy, can be coupled to the acoustic levitator so that chemical information can be
simultaneously provided (Brotton and Kaiser, 2013).
Acoustic levitation has been used in a variety of research fields to investigate interactions
of single solid/liquid particles with different gases (Kavouras and Krammer, 2003a; Mason et
al., 2008; Schenk et al., 2012), including water vapor. For example, Schenk et al. (2012) used
an acoustic levitator to measure hygroscopicity of imidazolium-based ionic liquids, and low
temperature acoustic levitation was developed to study homogeneous and heterogeneous
freezing of aqueous droplets (Ettner et al., 2004; Diehl et al., 2009; Diehl et al., 2014). Particles
which can be acoustically levitated are typically >20 μm (Mason et al., 2008; Krieger et al.,
2012), while most of atmospheric aerosol particles are significantly smaller (Seinfeld and
Pandis, 2016). Therefore, compared to the other two levitation techniques (i.e. EDB and optical
levitation), acoustic levitation is much less widely utilized in atmospheric chemistry (Krieger
et al., 2012).





### 4.4 Discussion

Both EDB and optical levitation can measure liquid water content for unsaturated and supersaturated samples, as particles used in these experiments are free of contact with other substances. EDB measures relative mass change to quantify aerosol liquid water content, and thus there is no constrain on particle shape; whereas for optical levitation, particle diameter change is usually measured optically, and particles under investigation need to be spherical. Both techniques may not be sensitive enough to study water adsorption. To our knowledge, they have not been used to investigate hygroscopic properties of ambient aerosol particles, though in principle they both have the capacity. One reason is that particles that can be explored using these techniques are usually one order of magnitude larger than those typically found in the troposphere. Another reason could be that only one particle can be examined in each experiment, while there are numerous aerosol particles in the ambient air. One unique advantage of these two techniques is that size, chemical composition and optical properties of levitated particles can be obtained in an online and noninvasive manner, making them very valuable to explore aerosol properties and processes at the fundamental level (Lee et al., 2008; Krieger et al., 2012).

### 5 Aerosol particles

In this section techniques that can be employed to investigate hygroscopic properties of airborne aerosol particles and can also be deployed for field measurements are reviewed. We discuss in Section 5.1 humidity-tandem differential mobility analysers which measure mobility diameter change of aerosol particles upon humidity change. Hygroscopic growth would further lead to change in aerosol optical properties, which can be measured to infer aerosol hygroscopicity, as reviewed in Section 5.2. In Section 5.3 we discuss in brief a few techniques developed to explore hygroscopic properties of black carbon aerosol in specific.



### 5.1 Humidity-tandem differential mobility analyser (H-TDMA)

### 5.1.1. Basic H-TDMA

The tandem differential mobility analyser (TDMA) was pioneered in 1978 and called the aerosol mobility chromatograph at that time (Liu et al., 1978). The terminology "TDMA" was first introduced in 1986 in a study (Rader and McMurry, 1986) which showed that size change as small as 1% could be readily measured. In addition to size change due to humidification (humidity-TDMA), TDMAs can also be used to measure particle size change due to other processing such as heating (Bilde et al., 2015a). H-TDMA is probably the most widely used technique for aerosol hygroscopicity measurement in both laboratory (Gibson et al., 2006; Herich et al., 2009; Koehler et al., 2009; Wex et al., 2009a; Good et al., 2010b; Wu et al., 2011; Hu et al., 2014; Lei et al., 2014; Gomez-Hernandez et al., 2016; Jing et al., 2016; Zieger et al., 2017) and field studies (McMurry and Stolzenburg, 1989; Swietlicki et al., 2008; Ye et al., 2011; Ye et al., 2013; Wang et al., 2014b; Yeung et al., 2014b; Atkinson et al., 2015; Cheung et al., 2015; Wu et al., 2016; Sorooshian et al., 2017). There are a number of H-TDMAs developed and used by individual research groups, and all the instruments follow the same principle. Recently these instruments have also become commercially available, e.g., from Brechtel Manufacturing Inc. (Lopez-Yglesias et al., 2014) and MSP Corporation (Sarangi et al., 2019). Swietlicki et al. (2008) provided a good description of the operation principle, and discussed potential error sources for H-TDMA measurements; Duplissy et al. (2009) analyzed the result from an intercomparison of six different H-TDMAs and recommended guidelines for design, calibration, operation and data analysis for H-TDMAs. Below we describe in brief how a typical H-TDMA works.

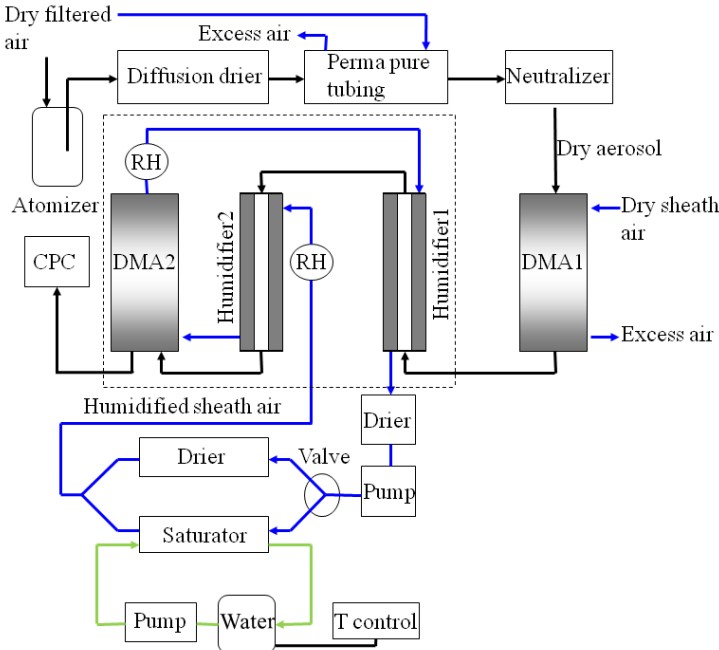


**Figure 21.** Schematic diagram of a typical H-TDMA instrument. Reprint with permission by

Jing et al. (2016). Copyright 2016 Copernicus Publications.


As illustrated in Fig. 21, polydisperse ambient or laboratory-generated aerosol particles
were sampled through an aerosol dryer to reduce the RH to <15%, and the dry aerosol flow
was passed through a neutralizer and then the first DMA (DMA1) to generate quasi-
monodisperse aerosol particles. After that, the aerosol flow was delivered through a
humidification section to be humidified to a given RH, and aerosol particles exiting the
humidification section were monitored using the second DMA (DMA2) coupled with a
condensation particle counter (CPC) to provide number size distributions. The growth factor
(GF) is defined as the ratio of aerosol mobility diameter at a given RH to that at dry condition.
The raw H-TDMA data should be inverted to retrieve the actual growth factor probability
density function (Rader and McMurry, 1986; Gysel et al., 2009; Good et al., 2010a), and
currently the inversion algorithm developed by Gysel et al. (2009) is widely used. One major



uncertainty for H-TDMA measurements stems from the accuracy of RH in the second DMA,
and considerable efforts are needed to minimize the RH and temperature fluctuation (Swietlicki
et al., 2008; Duplissy et al., 2009; Massling et al., 2011; Lopez-Yglesias et al., 2014). The
residence time in the humidification section should exceed 10 s for aerosol particles to reach
the equilibrium under a given RH, while it should not be more than 40 s due to potential
evaporation of semi-volatile species (Chan and Chan, 2005; Duplissy et al., 2009). In addition,
it is important to check the H-TDMA performance via comparing the measured GF with
theoretical values for reference aerosol particles, such as $(NH_4)_2SO_4$ and NaCl (Swietlicki et
al., 2008; Duplissy et al., 2009).
In typical laboratory studies (Herich et al., 2009; Koehler et al., 2009; Jing et al., 2016;
Zieger et al., 2017), aerosol size is measured as different RH using the H-TDMA to get the
RH-dependent GF. Humidograms, in which GF are plotted as a function of RH, are shown in
Fig. 22 for NaCl and synthetic sea salt aerosol particles, suggesting that at a given RH, GF of
sea salt aerosol is 8-15% smaller than NaCl aerosol (Zieger et al., 2017). Since both NaCl and
synthetic sea salt aerosol particles are non-spherical under dry conditions, growth factors were
reported after shape factor correction. The difference in GF between NaCl and synthetic sea
salt aerosols was attributed to the presence of hydrates (such as the hydrates of $MgCl_2$ and
$CaCl_2$) with lower hygroscopicity (when compared to NaCl) in synthetic sea salt (Zieger et al.,

2017).





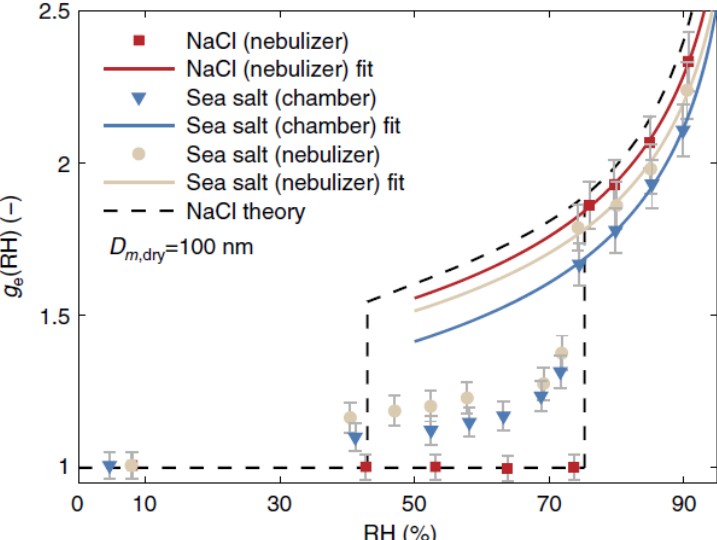


**Figure 22.** Measured hygroscopic growth factors of NaCl and synthetic sea salt aerosol

particles as different RH. NaCl aerosol particles were generated using a nebulizer, and both a

nebulizer and a sea spray chamber were used to generate sea salt aerosol particles. Reprint with

permission by Zieger et al. (2017). Copyright 2017 The Author(s).

H-TDMA has been widely used to investigate hygroscopic growth of secondary organic

aerosol (Prenni et al., 2007; Duplissy et al., 2008; Wex et al., 2009b; Good et al., 2010b;

Massoli et al., 2010; Duplissy et al., 2011; Alfarra et al., 2013; Zhao et al., 2016), which

significantly contributed to submicrometer aerosol particles over the globe (Zhang et al., 2007).

Using an aerosol flow tube, Massoli et al. (2010) generated secondary organic aerosols (SOA)

via OH oxidation of α-pinene, 1,3,5-trimethylbenzenen (TMB), m-xylene and a 50:50 mixture

of α-pinene and m-xylene, and measured their hygroscopic growth at 90% RH using a H-

TDMA. As shown in Fig. 23, measured GF at 90% RH ranged from 1.05 (non-hygroscopic) to

1.35 (moderately hygroscopic) for SOA systems examined, increasing linearly with O:C ratios

determined using an Aerodyne High Resolution Time-of-Flight Mass Spectrometer (Massoli



et al., 2010). In addition, for most SOA systems studied, single hygroscopicity parameters ($\kappa$)
derived from H-TDMA measurements were smaller than these derived from CCN activity
measurements (Massoli et al., 2010). Gaps between hygroscopic growth and cloud activation
have also been observed in a number of other studies for SOA (Prenni et al., 2007; Juranyi et
al., 2009; Petters et al., 2009; Wex et al., 2009b; Good et al., 2010b; Whitehead et al., 2014;
Zhao et al., 2016). One major reason for such gaps is that SOA usually contain substantial
amount of slightly soluble materials, which only undergo partial dissolution under water-
subsaturated conditions but can be dissolved to a significantly larger extent under water
supersaturated conditions (when more water is available). Further discussion on reconciliation
between hygroscopic growth and cloud activation can be found elsewhere (Petters et al., 2009;
Wex et al., 2009b). In another study (Li et al., 2014), H-TDMA was used to explore
hygroscopic properties of SOA formed via OH oxidation and direct photolysis of
methoxylphenol (a model compound for biomass burning aerosol) in the aqueous phase. For
SOA generated from aqueous phase OH oxidation, GF at 90% RH was observed to increase
linearly with the O:C ratio, but the slope was around three times smaller than that reported by
Massoli et al. (2010).

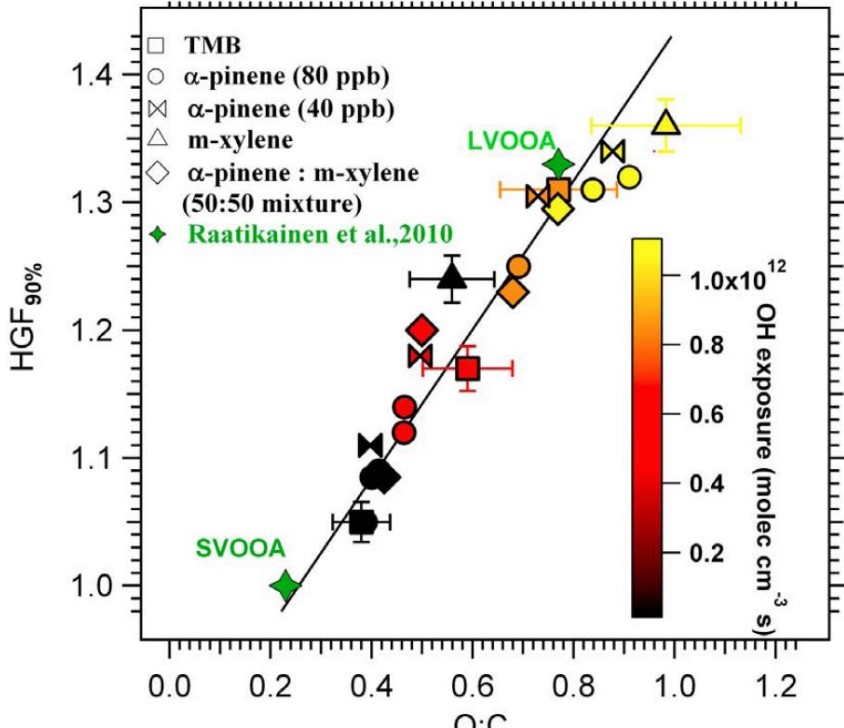


**Figure 23.** Growth factors of SOA measured using a H-TDMA at 90% RH as a function of

O:C ratios. Reprint with permission by Massoli et al. (2010). Copyright 2010 American

Geophysical Union.

Since RH scanning is time-consuming, in most ambient applications H-TDMA

measurements are usually carried out at a fixed RH (mostly 90%, and 85% to a less extent) for

one or a few dry particle diameters (Swietlicki et al., 2008; Kreidenweis and Asa-Awuku, 2014;

Cheung et al., 2015). Usually at least one diameter in the center of Aitken mode (~50 nm) and

one size in the center of the accumulation mode (~150 nm) are selected (Swietlicki et al., 2008).

The second DMA is typically scanned over a diameter range to cover a corresponding GF range

between 0.9 and 2.0 (sometimes up to 2.5) at 90% RH (Swietlicki et al., 2008). However, there

have been a few studies which measured GF of size-selected ambient aerosols as a function of



RH (Santarpia et al., 2004; Cheung et al., 2015). For example, Cheung et al. (2015) measured
the GF of ambient aerosol particles (100 and 200 nm) as a function of RH (10-93 %) in Hong
Kong using a H-TDMA, and found that the derived $\kappa$ values at (or above) 90% RH were
significantly larger than those derived at 40% RH. Each set of such measurements took ~3 h,
limiting its application to periods with large fluctuation in aerosol composition (Cheung et al.,

2015).

To further understand hygroscopic properties of ambient aerosol particles, aerosol

hygroscopicity closure studies have been widely carried out (Swietlicki et al., 1999; Dick et al.,
2000; Gysel et al., 2007; Cerully et al., 2011; Wu et al., 2013; Wu et al., 2016; Schurman et al.,
2017; Hong et al., 2018). In such studies, hygroscopic growth measurements using H-TDMA
are concurrently performed with aerosol chemical composition measurements, and measured
growth factors can then be compared to these calculated based measured chemical composition.
Aerosol chemical compositions were usually measured offline in the early stage (Swietlicki et
al., 1999; Dick et al., 2000) and have been increasingly determined online with high time
resolution using aerosol mass spectrometry (Gysel et al., 2007; Wu et al., 2013) and single
particle mass spectrometry (Wang et al., 2014c; Li et al., 2018a). For example, Wu et al. (2013)
used a H-TDMA to measure aerosol hygroscopic growth at 90% RH and an Aerodyne High
Resolution Time-of-Flight Mass Spectrometer (HR-ToF-AMS) to measure size-resolved
aerosol chemical composition at a middle-level mountain area in central Germany. Single
hygroscopicity parameters, $\kappa_{htdma}$, derived from growth factors measured using H-TDMA, were
compared to those derived from aerosol composition ($\kappa_{chem}$), assuming ideal mixing. If the
average compositions of submicron particles were used to calculate $\kappa_{chem}$, reasonably good
agreement between $\kappa_{htdma}$ and $\kappa_{chem}$ was found for 250 nm particles while no correlation was
observed for 100 nm particles (Wu et al., 2013). If size-resolved aerosol compositions were
used to calculate $\kappa_{chem}$, as shown in Fig. 24, good closure between $\kappa_{chem}$ and $\kappa_{htdma}$ were found




for all the four particle sizes. Fig. 24 also reveals that $\kappa_{chem}$ were significantly larger than $\kappa_{htdma}$,
indicating that ideal mixing assumption may overestimate aerosol hygroscopic growth (Wu et
al., 2013). Simultaneous H-TDMA and HR-ToF-AMS measurements were also carried out at
a coastal suburban site in Hong Kong (Yeung et al., 2014a). Approximations for growth factors
of organic aerosols, using the fraction of m/z 44, the oxygen-to-carbon ratio and PMF-resolved
organic factors from HR-ToF-AMS measurements, did not yield better closure results, likely
because of the overall dominance of sulfate during the whole measurement period.

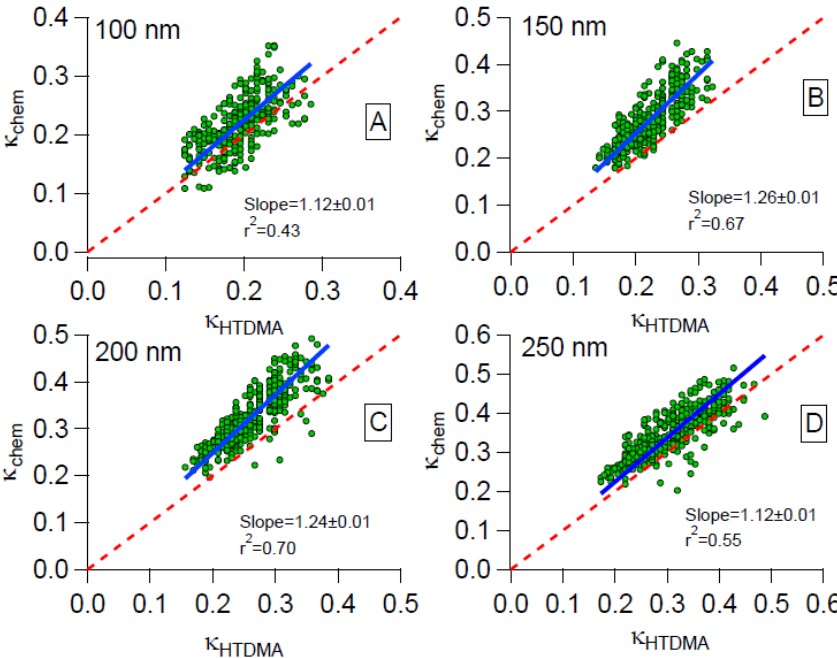


**Figure 24.** Comparison between $\kappa_{chem}$ (calculated using size-resolved aerosol compositions)
and $\kappa_{htdma}$ (derived from H-TDMA measurements) for aerosol particles with dry diameters of
(a) 100, (b) 150, (c) 200 and (d) 250 nm. Reprint with permission by Wu et al. (2013).
Copyright 2013 Copernicus Publications.




H-TDMA measurements in Shanghai at wintertime showed that aerosol particles (250 nm
in dry diameter) could be classified into two modes according to their hygroscopicity (Wang
et al., 2014b). The first mode had growth factors of ~1.05 at 85% RH, mainly containing fresh
elemental carbon and minerals, as revealed by measurements using an Aerosol Time-of –Flight
mass spectrometer. In contrast, the second mode had growth factors of ~1.46 at 85% RH and
were enriched with elemental carbon and organic carbon particles internally mixed with
secondary inorganic materials.
**5.1.2 H-TDMAs with extended performance**
While most H-TDMAs only work at around room temperature, Weingartner et al. (2002)
designed a H-TDMA which could measure hygroscopic growth of aerosol particle below 0 ºC
(temperature: -20 to 30 ºC; RH: 10-90 %). Measured hygroscopic growth factors showed good
agreement with theoretical calculations for $(NH_4)_2SO_4$, NaCl and $NaNO_3$ at both 20 and -10 ºC
(Gysel et al., 2002). This instrument was subsequently deployed at a high-alpine site (3580 m
above the seal level) to investigate hygroscopic properties of ambient aerosol particles at -10 ºC
(Weingartner et al., 2002), and the average GF at 85% RH were measured to be 1.44, 1.49 and
1.53 for aerosol particles with dry diameters of 50, 100 and 250 nm.
RH in the troposphere frequently exceeds 90%, and it is desirable to investigate
hygroscopic growth of aerosol particles at >90% RH. Hennig et al. (2005) developed a high
humidity TDMA which could be operated at 98% RH, and the absolute accuracy of RH at 98%
was ±1.2%. It was found that within the uncertainties, the measured GF in the RH range of 84-
98% agreed well with theoretical values (Hennig et al., 2005). The Leipzig Aerosol Cloud
Interaction Simulator (LACIS), a laminar flow tube designed to study cloud formation and
growth, could be operated at stable RH ranging from almost 0% up to 99.1% (Stratmann et al.,
2004), and aerosol particles and/or droplets exiting the flow tube were detected using an optical
particle sizer especially developed for this instrument. LACIS was employed to study



hygroscopic growth of $(NH_4)_2SO_4$ and NaCl aerosol particles at 85.8-99.1% RH (Wex et al.,
2005). At 99% RH, measured GF values agreed well with these predicted assuming solution
ideality for NaCl; whereas for $(NH_4)_2SO_4$, solution ideality assumption would overestimate GF
values by up to 20% (Wex et al., 2005). In a following study (Niedermeier et al., 2008), LACIS
was used to investigate hygroscopic growth of sea salt aerosol up to 99.1% RH.

Long duration is required by the second DMA to measure size distributions of humidified

aerosol particles, and therefore the H-TDMA technique is usually quite slow. It typically takes
~30 min for a traditional H-TDMA to determine GF values at a given RH for five different dry
diameters (Cerully et al., 2011; Pinterich et al., 2017b). Instruments with fast duty cycles are
of great interest and have been developed and deployed (Sorooshian et al., 2008; Pinterich et
al., 2017a; Pinterich et al., 2017b). For example, after replacing the second DMA (used in the
traditional H-TDMA) with a water-based fast integrated mobility spectrometer which could
provide 1 Hz size distribution measurements (Pinterich et al., 2017a), the improved instrument,
called the humidity-controlled fast integrated mobility spectrometer (HFIMS), only took ~3
min to measure GF of particles with five different dry diameters at a given RH (Pinterich et al.,
2017b).

Since the upper size limit is <1000 nm for a typical DMA and GF values at 90% RH can

be >2 for atmospheric particles, most H-TDMAs can only be used for particles with dry
diameters smaller than 500 nm (McFiggans et al., 2006; Swietlicki et al., 2008). Several
instruments, which could measure hygroscopic growth of aerosol particles larger than 500 nm
in dry diameter, have been developed (Kreisberg et al., 2001; Hegg et al., 2007; Massling et
al., 2007; Snider and Petters, 2008; Kaaden et al., 2009; Kim et al., 2014). One obvious
approach to overcome the DMA sizing limit is to use optical particle counters for particle sizing,
as adopted by some previous studies (Kreisberg et al., 2001; Hegg et al., 2007; Snider and
Petters, 2008). Another approach is to use Aerodynamic Particle Sizers (APS) for particle


sizing (Massling et al., 2007; Kaaden et al., 2009; Schladitz et al., 2011; Kim and Park, 2012).
For example, a H-DMA-APS was developed to explore hygroscopic growth of large aerosol
particles (Massling et al., 2007; Kaaden et al., 2009). As shown in Fig. 25, the dry aerosol flow
was first delivered through a custom-built high aerosol flow-DMA (HAF-DMA) which could
select particles with dry mobility diameters over 1000 nm, and the dry aerosol flow exiting the
DMA was split into two identical flows; the first flow was directly sampled by the first APS to
measure the aerodynamic size distribution under dry conditions, and the second flow was first
delivered through a humidifier to be humidified to a given RH (e.g., 90%) and then sampled
into the second APS so that the aerodynamic size distribution of the humidified aerosol was
measured.

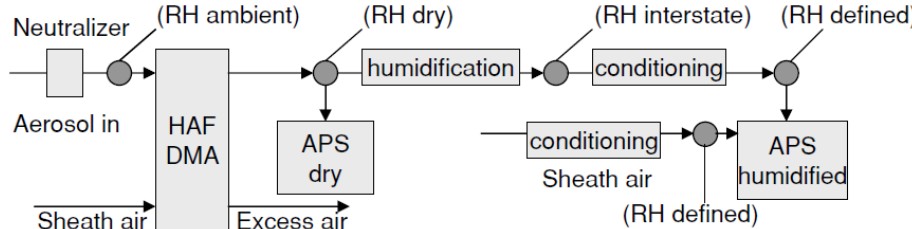


**Figure 25.** Schematic diagram of a H-DMA-APS apparatus. Reprint with permission by
Kaaden et al. (2009). Copyright 2009 Blackwell Munksgaard.

The utilization of H-TDMAs to measure aerosol hygroscopic growth factors assumes

particle sphericity. Some particles in the atmosphere, such as mineral dust and soot, are known
to be non-spherical, and therefore GF measured using H-TDMA may not correctly reflect the
amount of aerosol liquid water (Weingartner et al., 1997; Rissler et al., 2005; Vlasenko et al.,
2005; Koehler et al., 2009; Tritscher et al., 2011). Very recently an instrument, called
differential mobility analyser-humidified centrifugal particles mass analyser (DMA-HCPMA),
was developed to measure mass change of submicron aerosol particles at different RH (10-
95 %) (Vlasenko et al., 2017). In this set-up, a dry aerosol flow was delivered through a DMA



to produce quasi-monodisperse particles and then through an aerosol humidifier to be
humidified to a give RH; after that, the aerosol flow was delivered through a centrifugal particle
mass analyser (which would classify aerosol particles according to their mass-to-charge ratios)
(Olfert and Collings, 2005; Rissler et al., 2014; Kuwata, 2015) and then a CPC so that aerosol
particle mass could be determined as a function of RH (Vlasenko et al., 2017). The measured
mass growth factors were found to agree well with theoretical values for $(NH_4)_2SO_4$ and NaCl,
and this newly-developed DMA-HCPMA set-up was successfully deployed to explore
hygroscopic properties of ambient aerosol particles (Vlasenko et al., 2017). It can be expected
that DMA-HCPMA would significantly improve our knowledge of hygroscopicity of non-
spherical aerosol particles.
**5.2 Optical properties**

Optical properties of aerosol particles depend on their size and refractive indices, both

strongly affected by their hygroscopic properties. Measurements of aerosol optical properties
as a function of RH, indispensable for elucidating the impacts of aerosol particles on visibility
and radiative balance, can be used to infer aerosol hygroscopicity. Several techniques have
been developed and deployed, as discussed in this section.
**5.2.1 Extinction**

Cavity ring-down spectroscopy (CRDS), a highly sensitive method for optical extinction

measurement, has been extensively employed for gas and aerosol detection (Brown, 2003;
Baynard et al., 2006; Baynard et al., 2007; Langridge et al., 2011; Sobanski et al., 2016; Peng
et al., 2018). For a typical CRDS set-up, a laser beam pulse is coupled into a high-finesse
optical cavity (which has one high reflectivity mirror on each end) from one end of the cavity,
and the decay of the intensity of the light transmitted from the other end is monitored. The
change in decay lifetimes of transmitted light intensity can be related to the extinction
coefficient, $\alpha_{ext}$, using Eq. (3) (Baynard et al., 2007; Langridge et al., 2011):



$$\alpha_{ext} = \frac{R_L}{c}\left(\frac{1}{\tau} - \frac{1}{\tau_0}\right) \quad (3)$$

where $R_L$ is the ratio of the distance between the two mirrors to the length of the cavity filled
with aerosol particles, $c$ is the speed of light (m s$^{-1}$), and $\tau$ and $\tau_0$ are the measured decay
lifetimes of light intensity with and without aerosol particles present in the cavity. If aerosol
particles delivered into the cavity are monodisperse, the extinction coefficient of each
individual particles, $\sigma_{ext}$, can be calculated using Eq. (4) (Freedman et al., 2009):
$$\sigma_{ext} = \frac{\sigma_{ext}}{N_p} \quad (4)$$

where $N_p$ is the aerosol number concentration (cm$^{-3}$).

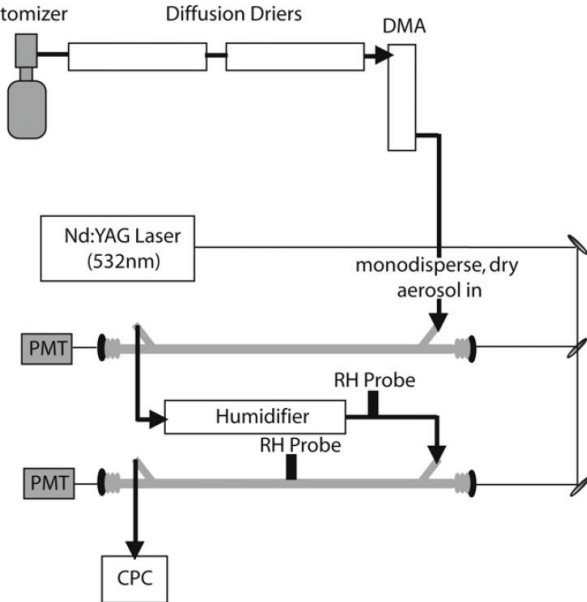


**Figure 26.** Schematic diagram of the apparatus used by Tolbert and co-workers to measure the
dependence of aerosol light extinction on RH. Reprint with permission by Beaver et al. (2008).
Copyright 2008 IOP Publishing Ltd.

A CRD spectrometer was employed by Tolbert and co-workers to investigate the effects

of RH on aerosol optical extinction at 532 nm, and its schematic diagram is depicted in Fig. 26



(Beaver et al., 2008). The aerosol flow generated using an atomizer was delivered through
diffusion dryers to reduce its RH to <10% and passed through a DMA to produce quasi-
monodisperse aerosol particles. The aerosol flow was then delivered into the first cavity to
measure the aerosol optical extinction at 532 nm under dry conditions; after that, the aerosol
flow entered a humidifier to be humidified to a given RH, and was then delivered into the
second cavity to measure the aerosol optical extinction under the humidified condition. In the
final, the aerosol flow was sampled by a CPC to measure the number concentration. For
$(NH_4)_2SO_4$ aerosol particles in the size range of 200-700 nm, the measured optical growth
factors at 80% RH, defined as the ratio of the extinction coefficient at 80% RH to that under
dry conditions, were found to be in good agreement with those calculated from diameter-based
growth factors using the Mie theory (Garland et al., 2007).

CRDS was used to examine the effect of RH on aerosol optical extinction for phthalic

acid, pyromellitic acid and 4-hydroxybenzoic acid aerosol particles in the size range of 150-
500 nm (Beaver et al., 2008). The optical growth factors were found to be smaller for the three
organic compounds examined, compared to $(NH_4)_2SO_4$. For example, for aerosol particles with
a dry diameter of 335 nm, optical growth factors at 80% RH were measured to be 1.3 and 1.1
for phthalic and pyromellitic acid (Beaver et al., 2008), compared to 3.0 for $(NH_4)_2SO_4$. Optical
extinction coefficients of 4-hydroxybenzoic acid particles at 80% RH were smaller than those
under dry conditions (Beaver et al., 2008), implying that morphological and structural change
may occur for these particles during humidification. Similarly, optical growth factors of illite
and kaolinite aerosol particles were found to be <1 at 50 and 68% RH (Attwood and Greenslade,
2011), due to structural rearrangement of clay mineral particles after water uptake. Optical
growth factors of internally mixed aerosol particles, which contained $(NH_4)_2SO_4$ and organic
materials, were also studied (Garland et al., 2007; Robinson et al., 2013; Robinson et al., 2014).
Another study (Flores et al., 2012) measured optical growth factors (at wavelengths of 355 and





532 nm) at 80 and 90% RH for aerosol particles with different extent of optical absorption
ranging from purely scattering (e.g., $(NH_4)_2SO_4$) to highly absorbing (e.g., nigrosine), and
found good agreement between measured optical growth factors and those calculated using the
Mie theory.

CRDS has also been widely deployed to investigate optical extinction of ambient aerosol

particles at different RH (Zhang et al., 2014b; Atkinson et al., 2015; Brock et al., 2016a). For
example, an eight-channel CRD spectrometer was developed by NOAA Earth System
Research Laboratory (Langridge et al., 2011). This instrument could measure aerosol optical
growth factors at three wavelengths (405, 532 and 662 nm) simultaneously, and has been
successfully deployed for aircraft measurements (Langridge et al., 2011).

In addition to CRDS, broadband cavity enhanced spectroscopy (BBCEAS), also called

cavity enhanced differential optical absorption spectroscopy (CE-DOAS), is an alternative
high-finesse cavity based technique with high sensitivity in optical extinction measurements
(Platt et al., 2009; Washenfelder et al., 2013; Washenfelder et al., 2016). Compared to CRDS,
one major advantage of BBCEAS is that optical extinction can be measured as a function of
wavelength. BBCEAS, as described in details elsewhere (Platt et al., 2009; Varma et al., 2013;
Washenfelder et al., 2013; Zhao et al., 2014; Washenfelder et al., 2016; Wang et al., 2017a; Li
et al., 2018b), has also been widely used in gas and aerosol measurements. Zhao et al. (2014)
utilized BBCEAS to measure aerosol optical extinction at 641 nm as a function of RH, and for
200 nm $(NH_4)_2SO_4$, the measured optical growth factors agreed well with those calculated
using the Mie theory. The instrument was further deployed to simultaneously measure optical
extinction of ambient submicrometer aerosol at <20% and 85% RH at Hefei Radiation
Observatory. The result is displayed in Fig. 27, suggesting that the optical growth factors at 85%
RH varied from ~1 to >2.5 during the campaign (Zhao et al., 2017).





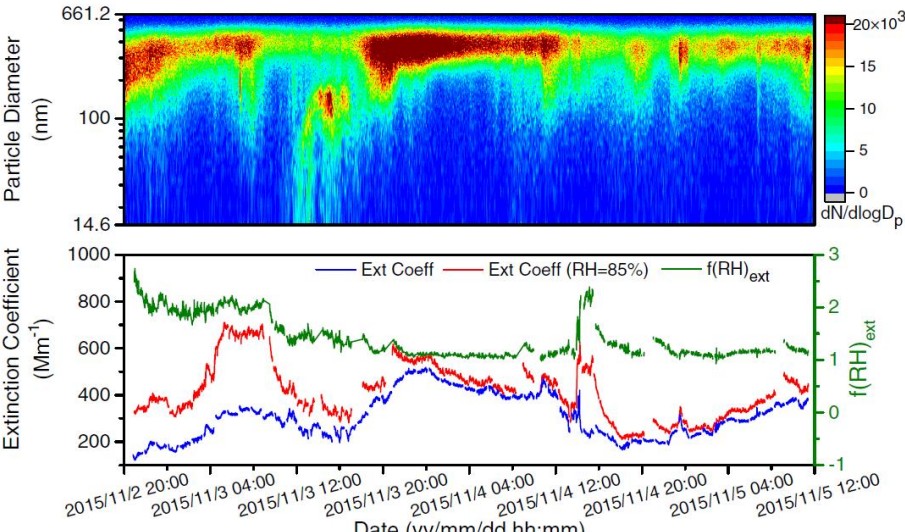

**Figure 27.** Aerosol properties measured at Hefei Radiation Observatory. Upper panel: aerosol number size distribution of submicrometer particles; lower panel: extinction coefficient of submicrometer particles under dry conditions (blue curve, left *y*-axis) and at 85% RH (red curve, left *y*-axis) and optical growth factors at 85% RH (green curve, right *y*-axis). Reprint with permission by Zhao et al. (2017). Copyright 2017 Optical Society of America.

**5.2.2 Scattering**

Humidified nephelometry, which was first developed as early as in the 1960s (Pilat and Charlson, 1966; Covert et al., 1972), has been widely used to measure aerosol light scattering coefficients at different RH (Rood et al., 1985; Carrico et al., 1998; Li-Jones et al., 1998; Day et al., 2000; Malm et al., 2000a; Malm et al., 2000b; Koloutsou-Vakakis et al., 2001; Fierz-Schmidhauser et al., 2010a; Zieger et al., 2010; Zieger et al., 2013; Kreidenweis and Asa-Awuku, 2014; Zhang et al., 2015; Titos et al., 2016). Due to its high time resolution, this technique is very suitable for online measurement of ambient aerosols. A very recent review paper (Titos et al., 2016) summarized and discussed theories, history, measurement uncertainties and ambient applications of this technique in a comprehensive manner. As a result,




herein we only introduce in brief its basic principle, representative instrumental configurations
and exemplary applications.

The scattering enhancement factor, $f$(RH), defined as the ratio of the aerosol scattering

coefficient at a given RH to that at dry conditions, is typically reported by humidified
nephelometry measurements (Kreidenweis and Asa-Awuku, 2014; Titos et al., 2016). Fig. 28
shows the schematic diagram of a humidified three-wavelength integrating nephelometer (TSI
3563) at 450, 550 and 700 nm (Fierz-Schmidhauser et al., 2010c). The aerosol flow was first
delivered through an aerosol humidifier which could increase the RH to 95% and then through
an aerosol dryer to reduce the RH to below 40%. After that, the aerosol flow was sampled into
the nephelometer to measure aerosol scattering coefficients at three different wavelengths. The
flow exiting the nephelometer was pulled through a mass flow controller (to control the sample
flow rate) by a pump. The performance of the aerosol dryer could be adjusted to vary the RH
of the flow entering the nephelometer, and thus scattering coefficients could be measured as a
function of RH (40-90 %); in addition, using such a configuration, light scattering properties
of supersaturated aerosol particles, i.e. the hysteresis effect, could be examined (Fierz-
Schmidhauser et al., 2010c). The humidified nephelometer was used to measure light scattering
properties of monodisperse $(NH_4)_2SO_4$ and NaCl aerosol particles with dry diameters of 100,
150, 240 and 300 nm, and the measured $f$(RH) values agreed with these predicted using Mie
theory (Fierz-Schmidhauser et al., 2010c). Some instruments could measure aerosol light
scattering at different RH in a simultaneous manner, via using two or more nephelometers in
parallel (Carrico et al., 1998).



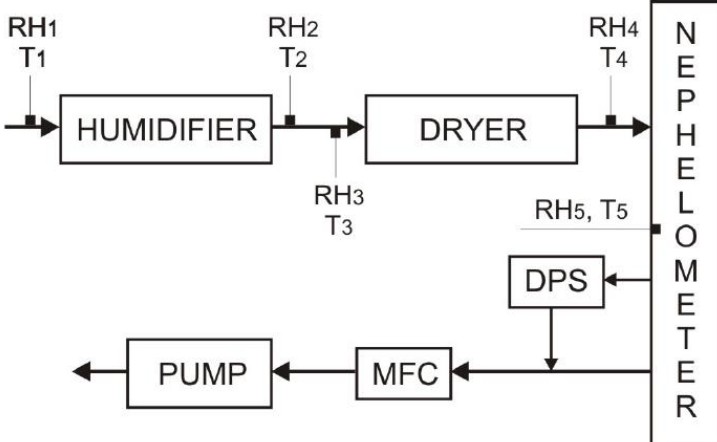


**Figure 28.** Schematic diagram of a humidified three-wavelength integrating nephelometer (DPS: dew point sensor; MFC: mass flow controller). Reprint with permission by Schmidhauser et al. (2010c). Copyright 2010 Copernicus Publications.


A number of previous studies have carried out field measurements of $f$(RH) at various locations over the globe (Zieger et al., 2013; Kreidenweis and Asa-Awuku, 2014; Titos et al., 2016). As summarized by Titos et al. (2016), $f$(RH) values (for 80-85% RH) were larger for marine sites (ranging from 1.5 to 3.5), when compared with most continental sites; furthermore, $f$(RH) values were found to be in the range of 1.1-2.1 for dust particles, and larger $f$(RH) values observed for dust may be caused by the co-presence of sea salt aerosol. A field study (Li-Jones et al., 1998) carried out on Barbados (West Indies) found that $f$(RH) values (for RH in the range of 67-83%) were very small (1.0-1.1) for mineral dust transported from North Africa, indicating that large variation in ambient RH may not lead to significant change in optical properties of mineral dust aerosol.

Since aerosol light scattering coefficients depend on particle size and refractive index in a complex manner even for spherical particles, it is not straightforward to link $f$(RH) with the aerosol liquid water content (Kreidenweis and Asa-Awuku, 2014). A number of studies (Malm



1789 and Day, 2001; Fierz-Schmidhauser et al., 2010b; Zieger et al., 2010; Chen et al., 2014;

1790 Kreidenweis and Asa-Awuku, 2014; Kuang et al., 2017; Kuang et al., 2018) have discussed

1791 how measured $f$(RH) values could be used to derive single hygroscopicity parameters ($\kappa$)

1792 (Petters and Kreidenweis, 2007) and aerosol liquid water contents. In addition, it should be

1793 emphasized that humidity-dependent aerosol scattering coefficients (as well as aerosol

1794 extinction and absorption coefficients) themselves are important parameters to assess the

1795 impacts of aerosols on visibility and direct radiative forcing.

1796 **5.2.3 Absorption**

1797  Photoacoustic spectroscopy has been developed and deployed to measure aerosol optical

1798 absorption in a direct manner (Arnott et al., 2003; Lack et al., 2009; Lewis et al., 2009;

1799 Moosmuller et al., 2009; Gyawali et al., 2012; Langridge et al., 2013; Lack et al., 2014). In

1800 brief, the aerosol flow is continuously sampled into a cell which serves as an acoustic resonator

1801 section and illuminated by a modulated laser beam. The laser radiation absorbed by aerosol

1802 particles is transferred to the surrounding air as heat, leading to the generation of acoustic wave

1803 which is amplified in the resonator and detected using a microphone (Moosmuller et al., 2009;

1804 Gyawali et al., 2012). The signal intensity measured by the microphone is proportional to

1805 optical absorption and can be used to derive aerosol optical absorption coefficients after proper

1806 calibration (Moosmuller et al., 2009; Gyawali et al., 2012). In principle, hygroscopic growth

1807 of aerosol particles at elevated RH would lead to increase in particle size and thus enhancement

1808 in aerosol optical absorption due to the lensing effect (Lewis et al., 2009). Nevertheless, several

1809 studies suggested that photoacoustic spectroscopy measurements at high RH are likely to

1810 significantly underestimate the actual aerosol optical absorption (Arnott et al., 2003; Lewis et

1811 al., 2009; Langridge et al., 2013). For example, Langridge et al. (2013) used photoacoustic

1812 spectroscopy at 532 nm to measure optical absorption of several types of aerosol particles with

1813 various hygroscopicity, morphology and refractive indices, and found that the measured



absorption exhibited strong low biases at high RH. The underestimation of optical absorption
is due to that acoustic signals are affected by evaporation of aerosol liquid water when aerosol
particles absorb radiation and get heated. As a result, Langridge et al. (2013) concluded that
photoacoustic spectroscopy was not a suitable technique to measure aerosol optical absorption
at elevated RH. Similarly, other techniques used for direct measurement of aerosol optical
absorption, such as the filter-based method and photothermal interferometry, did not perform
well at elevated RH either (Schmid et al., 2006; Sedlacek and Lee, 2007).
An indirect method has been developed (Khalizov et al., 2009; Xue et al., 2009; Brem et
al., 2012; Chen et al., 2015) to explore the effect of RH on aerosol optical absorption, which
was calculated as the difference between aerosol light extinction and scattering. In the set-up
developed by Brem et al. (2012), aerosol light extinction and scattering at three wavelengths
(467, 530 and 660 nm) were measured at different RH using an optical extinction cell and a
nephelometer. As RH was increased from 38 to 95%, light absorption of nigrosine aerosol was
enhanced by a factor of ~1.24 for all the three wavelengths (Brem et al., 2012). In some other
work (Khalizov et al., 2009; Xue et al., 2009; Chen et al., 2015), CRDS, instead of the optical
extinction cell, was used to measure the aerosol optical extinction.
**5.3 Other aerosol-based techniques**
Black carbon (BC) aerosol is of great concern due to its impacts on human health and
climate (Bond et al., 2013). The hygroscopicity of BC, varying with atmospheric aging
processes, largely determines its dry and wet deposition rates and thus lifetimes (Schwarz et
al., 2010; Wang et al., 2014a) and also affects its optical absorption through lensing effects
(Redemann et al., 2001). Therefore, it is important to understand hygroscopic properties of BC
aerosol in the troposphere; however, techniques discussed in Sections 5.1-5.2 are not specific
to BC-containing particles. Since typical BC mass fractions in submicrometer particles are only



a few percentages, in general these techniques cannot provide specific information on ambient
BC aerosol hygroscopicity.
Single particle soot photometers (SP2), as described in a number of studies (Gao et al.,
2007; Slowik et al., 2007; Schwarz et al., 2008; Moteki and Kondo, 2010), have been widely
employed to measure mass and mixing state of individual BC particles in the troposphere. In
brief, when an aerosol particle which contains a detectable amount of refractory BC enters a
SP2, it is heated by a laser beam (1064 nm) to the incandescence temperature, leading to the
emission of thermal radiation. The intensity of the thermal radiation, proportional to the mass
of refractory BC, is monitored to quantify the amount of BC contained by individual particles.
In addition, measurement of the light scattered by the particle during its initial interaction with
the laser beam can be used to derive the optical diameter. Therefore, a SP2 measures both the
mass of non-refectory BC and the optical diameter of each individual particles. In the last
several year a few SP2-based instruments have been developed to measure hygroscopic
properties of BC aerosol in specific (McMeeking et al., 2011; Liu et al., 2013a; Schwarz et al.,
2015; Ohata et al., 2016), as introduced below.
A SP2 was coupled to a H-TDMA to measure hygroscopic properties of BC aerosol
(McMeeking et al., 2011), and the experimental diagram is displayed in Fig. 29. The aerosol
flow was dried to <20% RH and then passed through the first DMA to produce quasi-
monodisperse aerosol with a specific size; after that, the aerosol flow was humidified to a
specific RH and then passed through the second DMA. The aerosol flow exiting the second
DMA was then split to two flows, sampled by a CPC and a SP2, respectively. The usage of
SP2 enabled identification of BC aerosol particles, and mobility diameter changes of aerosol
particles identified to be BC could be used to calculate hygroscopic growth factors specific to
BC aerosol; alternatively, hygroscopic properties of BC aerosol could be obtained from the
change in optical diameter measured by the SP2 (McMeeking et al., 2011). The H-TDMA-SP2





apparatus was deployed to investigate hygroscopic properties of BC aerosol in June-July 2011
at the Weybourne Atmospheric Observatory near the North Norfolk coastline. During this
campaign two types of BC aerosol with distinctive hygroscopicity were observed (Liu et al.,
2013a). Hygroscopic growth factors at 90% RH were measured to be ~1.05 for the first type
BC aerosol and ranged from ~1.25 to ~1.6 for the second type, depending on the composition
of soluble materials associated with BC particles (Liu et al., 2013a).

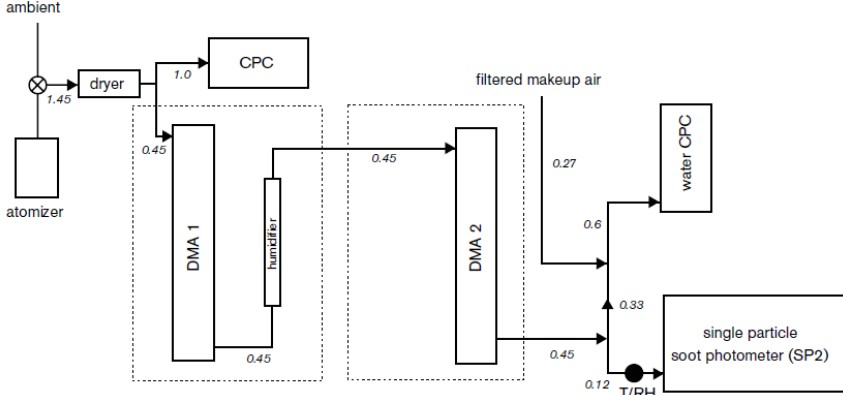


**Figure 29.** Schematic diagram of the H-TDMA-SP2 apparatus. Flow rates shown in this figure
are in the unit of L/min. Reprint with permission by McMeeking et al. (2011). Copyright 2011
Copernicus Publications.

Schwarz et al. (2015) developed a humidified-dual SP2 setup (HD-SP2) to measure

hygroscopic properties of BC aerosol. In this set-up, one sample flow was dried, and optical
diameters of each BC-containing particles were measured under dry conditions using the first
SP2; the other sample flow was first humidified to a given RH (e.g., 90%), and optical
diameters of individual BC-containing particles were determined using the second SP2. Optical
diameters of BC particles measured under dry and humidified conditions could then be used to
determine hygroscopic properties specific to BC-containing particles. The HD-SP2 was
deployed on the NASA DC-8 aircraft in the summer of 2013 to investigate hygroscopic



properties of BC aerosol in North American wildfire plumes (Perring et al., 2017). An average
$\kappa$ value of 0.04 was found for the sampled BC aerosol, and was increased by ~0.06 after 40 h
aging in the atmosphere (Perring et al., 2017).
In another study (Ohata et al., 2016), an aerosol particle mass analyser (APM) was coupled
to a humidified SP2 to investigate hygroscopic properties of BC aerosol. The experimental
scheme employed can be summarized as below (Ohata et al., 2016): (i) the sample flow, dried
to <10% RH, was delivered through an APM to select particles with a given mass-to-charge
ratio (with identical mass if multiple charged particles were excluded in data analysis); (ii) the
aerosol flow exiting the APM was humidified to a given RH and sampled into a SP2 to measure
optical diameters of BC-containing particles under humidified conditions. Since dry diameters
of BC-containing particles could be calculated from the mass of particles selected using the
APM, hygroscopic growth factors of BC aerosol could be consequently determined (Ohata et
al., 2016).
**5.4 Discussion**
All the techniques covered in Section 5 can be (and have been) used in laboratory and field
measurements. Since airborne particles are examined, aerosol water contents can be quantified
for unsaturated and supersaturated samples using these techniques. Because these techniques
rely on measurements of particle diameters to investigate hygroscopic properties, it can be non-
trivial to determine aerosol liquid water content for nonspherical aerosol particles. In addition,
they may not be sensitive enough to study water adsorption. Although in general these
techniques do not measure chemical compositions themselves, a number of offline and online
instruments, including advanced mass spectroscopic tools (e.g., aerosol mass spectrometers
and single particle mass spectrometers), are available to provide chemical information in
parallel, significantly deepening our knowledge of hygroscopic properties of complex aerosols.



## 6 Summary and final remarks

Hygroscopicity is one of the most important physiochemical properties of atmospheric aerosols, largely determining their environmental and climatic impacts. In addition to atmospheric science, it is also of great concern in many other scientific and technical fields, such as surface science, heterogeneous catalysis, geochemistry/astrochemistry, pharmaceutical and food science, and etc. A myriad of experimental techniques have been developed and employed to explore hygroscopic properties of aerosol particles for RH <100%. In this paper we have reviewed experimental techniques for investigating aerosol hygroscopicity in a comprehensive manner. Future directions are outlined and discussed below in order to improve existing techniques and develop new techniques for a better understanding of aerosol hygroscopicity.

1) The majority of instruments covered in this paper are not applicable to ambient aerosol particles. Future directions should focus on the development of aerosol hygroscopicity techniques that are field deployable, robust, and automatic. Especially up to now most ambient measurements conducted were ground-based, and therefore instruments which have high time resolution to be deployed on aircrafts (Langridge et al., 2011; Pinterich et al., 2017b) are highly needed.

2) The maximum RH that many techniques/instruments can currently reach is usually around 90%, and recent studies have revealed the importance of hygroscopic growth measurements at RH very close to 100% (Wex et al., 2009b). Therefore, efforts should be made to improve these instruments so that they can be employed to investigate hygroscopic properties at very high RH (e.g., up to 99% RH).

3) Temperatures in the troposphere range from ~200 K to >300 K, and temperature has been found to have a profound effect on particle phase state and thus liquid water content. Nevertheless, most techniques available currently, especially those which investigate



hygroscopic properties of aerosol particles, can only be operated at around room temperature.
Further instrumental development, which would enable hygroscopic growth measurements at
lower temperatures, is warranted.
4) Most techniques are operated under ambient pressure, while many processes involved
aerosol particles are often carried out at pressures substantially lower than atmospheric pressure
(Zhao et al., 2009; Schilling and Winterer, 2014; Rosenberger et al., 2018). As a result, new
techniques that allow direct measurements of hygroscopic properties at lower pressure are
needed for better characterization of aerosol hygroscopicity under conditions with reduced
pressure. Such instruments would also be very valuable for characterizing aerosol particles at
high altitudes where the pressure is significantly lower than the ground level.
5) Aerosol hygroscopicity is a property that depends on chemical compositions and its
measurements can be affected by phase state and viscosity of the particles. Application of
multiple techniques to examine the same type of atmospherically relevant particles will deepen
our understanding of aerosol hygroscopicity. In addition, simultaneous measurements of
chemical composition and other physicochemical properties (e.g., particle phase state and
viscosity) of aerosol particles of different hygroscopicity can be very valuable.
6) As shown in this review paper, many instruments employed to probe aerosol
hygroscopicity are custom built; furthermore, even for the same type of instruments,
operational protocols may vary at different groups. Instrumental comparisons, proven to be a
good approach to validate instrumental performance and identify potential issues, have been
carried out for H-TDMAs (Duplissy et al., 2009; Massling et al., 2011), and similar
intercomparison should be performed for other techniques and instruments. Furthermore,
standardized procedures for calibration, operation, data analysis and quality assurance, if can
be formulated, would help increase data quality for aerosol hygroscopicity measurements.




**Data availability**

This is a review paper, and all the data used come from literature cited.

**Author contribution**

Mingjin Tang and Chak K Chan conceived and coordinated this paper; Mingjin Tang, Chak K

Chan, Yong Jie Li, Hang Su, Qingxin Ma and Zhijun Wu wrote the paper with contribution

from all the other coauthors.

**Competing interests**

The authors declare that they have no conflict of interest.

**Acknowledgement**

This work is financially supported by National Natural Science Foundation of China

(91644106, 91744204, 4167517, 41875142 and 91844301), the Chinese Academy of Sciences

international collaborative project (132744KYSB20160036), State Key Laboratory of Organic

Geochemistry (SKLOG2016-A05) and Guangdong Foundation for Program of Science and

Technology Research (2017B030314057). Mingjin Tang would like to thank the CAS Pioneer

Hundred Talents program for providing a starting grant. This is contribution no. IS-XXXX

from GIGCAS.





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
