# Peer review of "A review of experimental techniques for aerosol hygroscopicity studies"

_Atmospheric Chemistry and Physics, 2019_

## Referee Comment (RC1) · Anonymous Referee #1 · 21 Aug 2019

The work is important and of relevance to the readership. The authors present a review of hygroscopicity measurements as it pertains to the atmosphere and also orthogonal scientific fields (surface science, heterogeneous catalysis, geochemistry/astrochemistry, pharmaceutical and food science, etc). Thus the publication will be of interest to the ACP readership and other fields. The authors have done a good job to describe "non-conventional atmospheric" hygroscopicity measurements. That is they provide an overview of comprehensive laboratory techniques that due to time or spatial resolution is not necessarily applied to field studies. For instance, the spectroscopy section is quite thorough and provides information on the numerous spectroscopy techniques that have been applied in controlled laboratory settings. The review is useful in that it provides a current overview of the current state of technology for hygroscop-

icity. However, the paper does not include a review of the theoretical hygroscopicity equations (although, I do not think that this is the purpose of the manuscript). Furthermore, I was quite disappointed to find that CCN and IN techniques were not discussed at all and perhaps this omission should be reflected in the title. E.g. " A review of experimental techniques for unsaturated aerosol hygroscopicity studies"

Regardless of this disappointment, I highly recommend the work for publication in ACP. The work will be cited heavily in the future. The following are a few concepts and ideas that may strengthen and or clarify ideas in the manuscript. I sincerely encourage the authors to consider addressing these comments before eventual publication.

Comments.

1. The fluorescence spectroscopy section seems tangential to the hygroscopicity discussion. Much of this sections suggest that EDB is the actual technique and then fluorescence is used to measures the particle properties.

2. Time resolution of hygroscopicity measurements should also be discussed as a recommendation to improve measurements. The DASH-SP and HFIMS are the only fast resolution hygroscopicity measurements techniques currently used. Recent work by Wang et al (HFIM) should also be discussed.

3. The authors may consider discussing how advances in orthogonal fields may be of future importance to atmospheric measurements. For example, although not currently relevant to aerosol, the production of highly sensitive humidity sensors should be considered (e.g., Liang et al, 2018 ). The Dash-P and HFIMS, use faster sizing instrumentation however faster RH technology may also advance studies.

Minor Corrections

L83, aqueous particle becomes supersaturated?

L85. efflorescence is also kinetically controlled? How? Not clear how this statement is made.

L70 – Remove or define "and etc"

L81 – spelling "dehumification"

L95 - , insert word - leading to the formation OF two coexisting liquid"

L95 – change to : in one particle"

L120 – Remove "in specific,"

L189 – Change recently to recent

L385 – Change isotherm to isothermal

L1480 – "aerosol size is measured as" to "aerosol size is measured AT"

Additional References to consider

Wang, Yang, et al. "Retrieval of High Time Resolution Growth Factor Probability Density Function from a Humidity-controlled Fast Integrated Mobility Spectrometer." Aerosol Science and Technology just-accepted (2019): 1-18.

Liang, Jun-Ge, et al. "Thickness effects of aerosol deposited hygroscopic films on ultra-sensitive humidity sensors." Sensors and Actuators B: Chemical 265 (2018): 632-643.

---

## Referee Comment (RC2) · Defeng Zhao (Referee) · 22 Aug 2019

The comments are attached.

[Figure]

The manuscript "A review of experimental techniques for aerosol hygroscopicity studies" presents a comprehensive and systematic review of the techniques used to study hygroscopicity of aerosols. The experimental techniques are classified into four types, according to how samples are prepared. For each method, besides experimental techniques, typical applications of this method to aerosol hygroscopicity study are provided. Finally, the future direction to improve these techniques are suggested, including improving these methods to use in more variable ambient environment (high RH, low pressure, low T), conducting more instrument inter-comparisons and investigating other physicochemical properties of aerosol together with hygroscopicity.

A comprehensive review of the techniques used to study aerosol hygroscopicity is lacking up to now, to the best of my knowledge, although previous papers well summarizes some techniques, especially the HTDMA techniques (Duplissy et al., 2009) and techniques to study physicochemical properties in general (Ault and Axson, 2017). Therefore, this manuscript would be beneficial to ACP readers. The manuscript is well written and clearly organized. I recommend publication of this manuscript in ACP after a few minor comments are addressed.

1.  The authors discussed the advantages and disadvantages/problem of each technique. In the summary part, I suggest authors to add a table to summarize these features so that readers can get an overview and this could somehow work as a guideline when one reads a paper on aerosol hygroscopicity studied using a certain method and chooses a suitable technique in their research.

2.  Some studies on other physicochemical properties are discussed this manuscript. While most of them are relevant to the topic of study, some may not be the focus of this manuscript, such as lines 1102-1105, 1232-1236, 1380-1383, 1400-1403. Condensing these texts might be desirable.

    Also the lines 1042-1051 (and Fig 15, 16) discussed the application of Raman spectroscopy to study heterogeneous reaction. Since the application of Raman spectroscopy to hygroscopicity has been demonstrated earlier in the manuscript, I suggest omitting this part, especially considering the figures are not considered to be officially published yet.

3.  Line 1575, it might be worth noting that "Aerosol Time-of–Flight mass spectrometer" is a single particle mass spectrometer, e.g. specify it by adding the abbreviation.

4.  Some texts are underlined (such as line 620 and other part). Is this a typeset problem?

**Fig. 1.**

---

## Author Comment (AC1) · 26 Aug 2019

Comments by referees are in blue.

Our replies are in black.

Changes to the manuscript are highlighted in red both in here and in the revised manuscript.

The work is important and of relevance to the readership. The authors present a review of hygroscopicity measurements as it pertains to the atmosphere and also orthogonal scientific fields (surface science, heterogeneous catalysis, geochemistry/astrochemistry, pharmaceutical and food science, etc). Thus the publication will be of interest to the ACP readership and other fields. The authors have done a good job to describe "non-conventional atmospheric" hygroscopicity measurements. That is they provide an overview of comprehensive laboratory techniques that due to time or spatial resolution is not necessarily applied to field studies. For instance, the spectroscopy section is quite thorough and provides information on the numerous spectroscopy techniques that have been applied in controlled laboratory settings. The review is useful in that it provides a current overview of the current state of technology for hygroscopicity. However, the paper does not include a review of the theoretical hygroscopicity equations (although, I do not think that this is the purpose of the manuscript). Furthermore, I was quite disappointed to find that CCN and IN techniques were not discussed at all and perhaps this omission should be reflected in the title. E.g. "A review of experimental techniques for unsaturated aerosol hygroscopicity studies". Regardless of this disappointment, I highly recommend the work for publication in ACP. The work will be cited heavily in the future. The following are a few concepts and ideas that may strengthen and or clarify ideas in the manuscript. I sincerely encourage the authors to consider addressing these comments before eventual publication.

**Reply:** We would like to thank Ref #1 for his/her insightful comments as well as recommending our manuscript for final publication. We do not review theories related to aerosol hygroscopicity because our manuscript is focused on techniques for aerosol hygroscopicity measurements; similarly, we do not review techniques used for CCN and IN measurements. However, in the original manuscript we have referred readers to literature where aerosol hygroscopicity theories and techniques for CCN and IN measurements are reviewed. We feel that the title we use is proper, although we understand what the referee means. This is because hygroscopicity usually means interaction of water vapor with particles at <100% RH. We have also addressed all the other comments adequately in the revised manuscript, as detailed below.

Comments.

1. The fluorescence spectroscopy section seems tangential to the hygroscopicity discussion. Much of this sections suggest that EDB is the actual technique and then fluorescence is used to measures the particle properties.

**Reply:** It is true that fluorescence spectroscopy does not measure aerosol hygroscopicity; however, it provides information (for example, the ratio of solvated water to free water) closely related to hygroscopicity. Therefore, we chose to discuss this technique using two paragraphs.

2. Time resolution of hygroscopicity measurements should also be discussed as a recommendation to improve measurements. The DASH-SP and HFIMS are the only fast resolution hygroscopicity measurements techniques currently used. Recent work by Wang et al (HFIM) should also be discussed.

**Reply:** Indeed time resolution is an important aspect, and this has been discussed in the original manuscript (future direction #1, page 92). We would like to thank the referee for bringing the work by Wang et al. (2019) to our attention, and in the revised manuscript (page 95) the work have been cited in addition to that by Langridge et al. (2011) and Pinterich et al. (2017b) as examples of high time resolution instruments for aerosol hygroscopicity measurements.

3. The authors may consider discussing how advances in orthogonal fields may be of future importance to atmospheric measurements. For example, although not currently relevant to aerosol, the production of highly sensitive humidity sensors should be considered (e.g., Liang et al, 2018). The Dash-P and HFIMS, use faster sizing instrumentation however faster RH technology may also advance studies.

**Reply:** The advancement in RH measurements will definitely be very valuable. In the revised manuscript (page 95, future direction #2) we have added the following sentences to discuss this aspect: "Furthermore, currently RH measurements typically have an absolute uncertainty of 1% or larger, and uncertainties in RH measurement would affect hygroscopic growth factors reported at a given RH, especially for high RH at which growth factors are more sensitive to RH; therefore, advancement in RH measurements (Liang et al., 2018) will contribute to the improvement in aerosol hygroscopicity measurement techniques."

Minor Corrections

**Reply:** We would like to thank the referee for carefully reading our manuscript and pointing out those typos, as detailed below.

L83, aqueous particle becomes supersaturated?

**Reply:** It means that the aqueous particle becomes a supersaturated solution. In the revised manuscript (page 4) we have expanded this sentence to provide further clarification: "the aqueous particle would become supersaturated (i.e. the aqueous particle becomes a supersaturated solution)."

L85. efflorescence is also kinetically controlled? How? Not clear how this statement is made.

**Reply:** Here we mean that efflorescence not only depends on thermodynamics but also is kinetically controlled. In the revised manuscript (page 4) we have expanded this sentence to provide further explanation: "Therefore, efflorescence is also kinetically controlled (in addition to being thermodynamically controlled) and…"

L70 – Remove or define "and etc"

**Reply:** As suggested, in the revised manuscript we have deleted "and etc".

L81 – spelling "dehumification"

**Reply:** In the revised manuscript this typo has been corrected.

L95 - , insert word - leading to the formation OF two coexisting liquid"

**Reply:** Corrected.

L95 – change to : in one particle"

**Reply:** Corrected.

L120 – Remove "in specific,"

**Reply:** Removed.

L189 – Change recently to recent

**Reply:** We have checked the manuscript and here "recently" should be used.

L385 – Change isotherm to isothermal

**Reply:** Corrected.

L1480 – "aerosol size is measured as" to "aerosol size is measured AT"

**Reply:** Corrected.

Additional References to consider

Wang, Yang, et al. "Retrieval of High Time Resolution Growth Factor Probability Density Function from a Humidity-controlled Fast Integrated Mobility Spectrometer." Aerosol Science and Technology just-accepted (2019): 1-18.

Liang, Jun-Ge, et al. "Thickness effects of aerosol deposited hygroscopic films on ultrasensitive

humidity sensors." Sensors and Actuators B: Chemical 265 (2018): 632-643.

**Reply:** As suggested, in the revised manuscript we have discussed these two studies.

---

## Author Comment (AC2) · 26 Aug 2019

Comments by referees are in blue.

Our replies are in black.

Changes to the manuscript are highlighted in red both in here and in the revised manuscript.

The manuscript "A review of experimental techniques for aerosol hygroscopicity studies" presents a comprehensive and systematic review of the techniques used to study hygroscopicity of aerosols. The experimental techniques are classified into four types, according to how samples are prepared. For each method, besides experimental techniques, typical applications of this method to aerosol hygroscopicity study are provided. Finally, the future direction to improve these techniques are suggested, including improving these methods to use in more variable ambient environment (high RH, low pressure, low T), conducting more instrument inter-comparisons and investigating other physicochemical properties of aerosol together with hygroscopicity. A comprehensive review of the techniques used to study aerosol hygroscopicity is lacking up to now, to the best of my knowledge, although previous papers well summarizes some techniques, especially the HTDMA techniques (Duplissy et al., 2009) and techniques to study physicochemical properties in general (Ault and Axson, 2017). Therefore, this manuscript would be beneficial to ACP readers. The manuscript is well written and clearly organized. I recommend publication of this manuscript in ACP after a few minor comments are addressed.

**Reply:** We would like to thank ref #2 for his/her insightful comments as well as recommending our manuscript for final publication. We have addressed all the comments adequately in the revised manuscript, as detailed below.

1. The authors discussed the advantages and disadvantages/problem of each technique. In the summary part, I suggest authors to add a table to summarize these features so that readers can get an overview and this could somehow work as a guideline when one reads a paper on aerosol hygroscopicity studied using a certain method and choose a suitable technique in their research.

**Reply:** We fully agree with the referee. In the revised manuscript (page 90-94), we have added one table and a few paragraphs to summarize key futures of major techniques for aerosol hygroscopicity measurements.

2. Some studies on other physicochemical properties are discussed this manuscript. While most of them are relevant to the topic of study, some may not be the focus of this manuscript, such as line 1102-1105, 1232-1236, 1380-1383, 1400-1403. Condensing these texts might be desirable.

**Reply:** We agree that these contents are not directly related to aerosol hygroscopicity; however, they are intentionally included because we want to show that these techniques can also be used to investigate other physicochemical properties besides aerosol hygroscopicity. Therefore, we would like to keep them in the manuscript.

Also the lines 1042-1051 (and Fig 15, 16) discussed the application of Raman spectroscopy to study heterogeneous reaction. Since the application of Raman spectroscopy to hygroscopicity has been demonstrated earlier in the manuscript, I suggest omitting this part, especially considering the figures are not considered to be officially published yet.

**Reply:** As requested, in the revised manuscript we have removed Figs. 15-16 and related text, since this part has not yet been officially published.

3. Line 1575, it might be worth noting that "Aerosol Time-of-Flight mass spectrometer" is a single particle mass spectrometer, e.g. specify it by adding the abbreviation.

**Reply:** In the revised manuscript (page 75) we have modified this sentence to make it clear that this instrument is a single particle spectrometer: "as revealed by measurements using a single particle mass spectrometer (Aerosol Time-of –Flight mass spectrometer)." Since this term only appears once in our manuscript, it is not necessary to add its abbreviation.

4. Some texts are underlined (such as line 620 and other part). Is this a typeset problem?

**Reply:** Because there are ~30 figures in our manuscript, we underline the text when a figure is mentioned in the text (such as "As shown in Fig. 1,"). Underlines will be removed when we upload the document required by final publication after the manuscript is accepted.

---

## Author Comment (AC3) · 26 Aug 2019

The comment was uploaded in the form of a supplement:
https://www.atmos-chem-phys-discuss.net/acp-2019-398/acp-2019-398-AC3-supplement.pdf